# GREEDY LEARNING TO OPTIMIZE WITH CONVERGENCE GUARANTEES

## ABSTRACT

Learning to optimize is an approach that leverages training data to accelerate the solution of optimization problems. Many approaches use unrolling to parametrize the update step and learn optimal parameters. Although L2O has shown empirical advantages over classical optimization algorithms, memory restrictions often greatly limit the unroll length and learned algorithms usually do not provide convergence guarantees. In contrast, we introduce a novel method employing a greedy strategy that learns iteration-specific parameters by minimizing the function value at the next iteration. This enables training over significantly more iterations while maintaining constant GPU memory usage. We parameterize the update such that parameter learning corresponds to solving a convex optimization problem at each iteration. In particular, we explore preconditioned gradient descent with multiple parametrizations including a novel convolutional preconditioner. With our learned algorithm, convergence in the training set is proved even when the preconditioner is neither symmetric nor positive definite. Convergence on a class of unseen functions is also obtained, ensuring robust performance and generalization beyond the training data. We test our learned algorithms on two inverse problems, image deblurring and Computed Tomography, on which learned convolutional preconditioners demonstrate improved empirical performance over classical optimization algorithms such as Nesterov's Accelerated Gradient Method and the quasi-Newton method L-BFGS.

## 1 INTRODUCTION

We consider the optimization problem

$$\min_x f(x), \tag{1}$$

with the assumption that $f : \mathcal{X} \to \mathbb{R}$ is convex, $L$-smooth and bounded below, where $\mathcal{X}$ is a Hilbert space. Classic optimization methods are built in a theoretically justified manner, with guarantees on their performance and convergence properties. For example, Nesterov's Accelerated Gradient Method (NAG) (Nesterov, 1983) accelerates classical first-order algorithms using momentum. However, practitioners often concentrate on problems within a much smaller class. For example, in reconstructing images from blurred observations $y$ generated by a blurring operator $A$, one might minimize a function from the class:

$$\mathcal{F} = \left\{ f : \mathcal{X} \to \mathbb{R} : f(x) = \frac{1}{2}\|Ax - y\|^2 + \mathcal{S}(x), y \sim \mathcal{P}(\mathcal{Y}) \right\}, \tag{2}$$

where $\mathcal{S} : \mathcal{X} \to \mathbb{R}$ is a chosen regularizer and $\mathcal{P}(\mathcal{Y})$ is some probability distribution on $\mathcal{Y}$ detailing the observations $y$ of interest. Learning to optimize (L2O) uses data to learn how to minimize functions $f \in \mathcal{F}$ in a small number of iterations. Typically, the solution at each iteration $t$ is updated by a parametrized function $G_\theta : \mathcal{X} \times \mathcal{X} \to \mathcal{X}$ (i.e. the update rule) as dependent on parameters $\theta_t$ at iteration $t$ as

$$x_{t+1} = x_t - G_{\theta_t}(x_t, \nabla f(x_t)). \tag{3}$$

Unrolling algorithms (Monga et al., 2021) directly parametrize the update step as a neural network, often taking the previous iterates of the solution updates and the gradients as input arguments to the neural network. For some $T > 0$, the parameters $\theta = (\theta_0, \cdots, \theta_T)$ can be learned to minimise the

loss

$$L(\theta) = \mathop{\mathbb{E}}_{f \in \mathcal{F}} \left[ \sum_{t=1}^{T+1} f(x_t) \right]. \tag{4}$$

Learned optimization algorithms often lack convergence guarantees, including many that use RNNs (Andrychowicz et al., 2016; Metz et al., 2019) or Reinforcement learning (Li & Malik, 2016). Liu et al. (2023) consider methods of the form $x_{t+1} = x_t - G_t \nabla f(x_t) + b_t$, for $f : \mathbb{R}^n \to \mathbb{R}$, a diagonal matrix $G_t \in \mathbb{R}^{n \times n}$, and a vector $b_t \in \mathbb{R}^n$. The $G_t$ and $b_t$ are constructed using the outputs of neural networks. However, their method does not guarantee convergence to a minimizer.

Other approaches achieve provable convergence, which can be enforced with safeguarding (Heaton et al., 2023), or constructing convergent algorithms by learning parameters within a provably convergent set (Banert et al., 2020; 2024). Tan et al. (2023a;b) learn mirror maps using input-convex neural networks within the mirror descent optimization algorithm such that the algorithm is provably convergent. Lastly, Sucker et al. (2024) and Sambharya & Stellato (2024) consider applying the PAC-Bayes framework to L2O.

Unlike NAG, Newton's method accelerates convergence by applying the inverse Hessian to the gradient, which can be costly in practice. Quasi-Newton methods like BFGS (Nocedal & Wright, 2006b) approximate the Hessian, and L-BFGS (Liu & Nocedal, 1989) is used when BFGS is too memory-intensive. Similarly, we aim to accelerate the optimization by learning a preconditioner $G_t$ in the update $x_{t+1} = x_t - G_t \nabla f(x_t)$.

Adaptive algorithms improve optimization during use. For example, Armijo line-search (Armijo, 1966) seeks to find a good step size at each iteration, while methods like AdaGrad (Duchi et al., 2011) and optimal diagonal preconditioners (Qu et al., 2024) adapt preconditioners. Online optimization (Hazan et al., 2016), with methods such as Coin Betting (Orabona & Pál, 2016) and Adaptive Bound Optimization (McMahan & Streeter, 2010), offers a game-theoretic perspective to optimization.

## 1.1 CONTRIBUTIONS

Our paper contributes in the following ways:

- A novel approach to L2O that learns parameters at each iteration sequentially, using a greedy approach by minimizing the function value at the next iteration. This enables training over significantly more iterations while maintaining constant GPU memory usage: Section 3.

- Convergence in the training set is proved even when the preconditioner is neither symmetric nor positive definite: Section 4. Furthermore, convergence is proved on a class of unseen functions under certain conditions using soft constraints for parameter learning.

- Learning parameters is a convex optimization problem for 'linear parametrizations' of $G_t$, enabling training that is significantly faster, with closed-form solutions for least-squares functions: Section 5.

- A novel parametrization of $G_t$ as a convolution operator. At iteration $t$ we learn a convolutional kernel $\kappa_t$ such that $G_t x = \kappa_t * x$. This parametrization is shown to outperform Nesterov's Accelerated Gradient and L-BFGS on test data: Section 6.

In Section 6, we validate our learned algorithms on two inverse problems: image deblurring and Computed Tomography (CT). Inverse problems represent a crucial class of optimization problems that appear in important fields such as medical imaging and machine learning. Many such problems have an associated forward operator which is highly ill-conditioned, making them an ideal test for optimization algorithms.

## 2 NOTATION

Let $\mathcal{X}$ be a Hilbert space with corresponding field $\mathbb{R}$ and norm $\|\cdot\|$. A function $f : \mathcal{X} \to \mathbb{R}$ is convex if for all $x, y \in \mathcal{X}$ and for all $\alpha \in [0, 1]$ $f(\alpha x + (1 - \alpha)y) \leq \alpha f(x) + (1 - \alpha)f(y)$. A function $f : \mathcal{X} \to \mathbb{R}$ is $L$-smooth with parameter $L > 0$ if its gradient is Lipschitz continuous, i.e., if for all $x, y \in \mathcal{X}$, $\|\nabla f(x) - \nabla f(y)\| \leq L\|x - y\|$. A function $f : \mathcal{X} \to \mathbb{R}$ is bounded below if there exists some $M \in \mathbb{R}$ such that $f(x) \geq M$ for all $x \in \mathcal{X}$. We say that $f \in \mathcal{F}_L$ if $f$ is convex,

$L$-smooth, and bounded below. We assume that the Hilbert space $\mathcal{X}$ has dimension $\dim(\mathcal{X}) = n$ and, therefore, admits a finite orthonormal basis $\{e_1, \cdots, e_n\}$. For $x \in \mathcal{X}$ and $j \in \{1, \cdots, n\}$, define $[x]_j := \langle x, e_j \rangle$. For $x, y \in \mathcal{X}$, define the pointwise product $x \odot y$ by $[x \odot y]_j := [x]_j [y]_j$. For Hilbert spaces $\mathcal{X}$ and $\mathcal{Y}$, denote the space of linear operators from $\mathcal{X}$ to $\mathcal{Y}$ by $\mathcal{L}(\mathcal{X}, \mathcal{Y})$. If $\mathcal{Y} = \mathcal{X}$, we write $\mathcal{L}(\mathcal{X})$. For example, if $\mathcal{X} = \mathbb{R}^n$, $\mathcal{L}(\mathcal{X})$ is the space of $n \times n$ matrices. Denote the adjoint of $A \in \mathcal{L}(\mathcal{X}, \mathcal{Y})$ by $A^*$, meaning that for $x \in \mathcal{X}, y \in \mathcal{Y}, \langle Ax, y \rangle = \langle x, A^*y \rangle$. Denote by $I \in \mathcal{L}(\mathcal{X})$ the identity operator: $I(x) = x$ for all $x \in \mathcal{X}$.

# 3 GREEDY LEARNING TO OPTIMIZE OF PRECONDITIONED GRADIENT DESCENT

This section introduces the proposed method: greedy learning to optimize. Firstly, we introduce how we parametrize the optimization algorithm as preconditioned gradient descent. Next, we detail our training data and define a loss function with which we learn parameters. We then provide an algorithm of how parameters are learned sequentially using a greedy approach. Lastly, we show how our learned algorithm is applied to unseen functions.

At each iteration $t \in \{0, 1, 2, \cdots\}$, we parametrize the linear operator $G_t$ using a Hilbert space $\Theta$ and learn parameters $\theta_t \in \Theta$ in the update

$$x_{t+1} = x_t - G_{\theta_t} \nabla f(x_t). \tag{5}$$

The following propositions show that it is possible to obtain convergence after just one iteration of the update (5). Firstly, we show that it is possible to even when $G$ is a pointwise operator, i.e. $Gx := p \odot x$ for some $p \in \mathcal{X}$.

**Proposition 1.** *Assume that $f : \mathcal{X} \to \mathbb{R}$ is convex, continuously differentiable, and has a global minimum, and take any initial point $x_0 \in \mathcal{X}$. Then there exists $p \in \mathcal{X}$ such that, $x_0 - p \odot \nabla f(x_0) \in \arg\min_x f(x)$.*

While the pointwise parametrization obtains convergence after one iteration for one function, for an arbitrary linear operator $G \in \mathcal{L}(\mathcal{X})$, under certain conditions, one can obtain convergence after one iteration for multiple functions.

**Proposition 2.** *For $k \in \{1, \cdots, N\}$, assume that $f_k : \mathcal{X} \to \mathbb{R}$ is convex, continuously differentiable, and has a global minimum, with any initial point $x_k^0 \in \mathcal{X}$. Assume that the set of gradients $\{\nabla f_1(x_1^0), \cdots, \nabla f_N(x_N^0)\}$ is linearly independent. Then if $N \leq n$, there exists an operator $P \in \mathcal{L}(\mathcal{X})$ such that $x_k^0 - P\nabla f_k(x_k^0) \in \arg\min_x f_k(x)$, for all $k \in \{1, \cdots, N\}$.*

Propositions 1 and 2 motivate learning $\theta_t$ by considering the function values only at the next iteration. In order to learn the parameters $\theta_t$ for $t \in \{0, 1, 2, \cdots\}$, we use a training dataset of functions $\mathcal{T} := \{f_1, \cdots, f_N\}$, with $f_k \in \mathcal{F}_{L_k}$ for $k \in \{1, \cdots, N\}$, with corresponding initial points $\mathcal{X}_0 := \{x_1^0, \cdots, x_N^0\}$.

We consider learning parameters using a regularizer $R : \Theta \to \mathbb{R}$ so that undesirable properties are penalized. At iteration $t$, we solve the optimization problem

$$\theta_t \in \arg\min_\theta \left\{ g_{t,\lambda_t}(\theta) := \frac{1}{N} \sum_{k=1}^N f_k(x_k^t - G_\theta \nabla f_k(x_k^t)) + \lambda_t R(\theta) \right\}, \tag{6}$$

for some regularization parameter $\lambda_t \geq 0$, which is used to balance the importance of the regularizer. Such a strategy is greedy, as learning refers to tuning the parameters $\theta_t$ considering only the function values at the next iteration, $f_k(x_k^{t+1})$. The sequential training procedure for parameter learning is detailed in Algorithm 1. For unrolling with a standard implementation of backpropagation, GPU memory requirements scale linearly with the number of training iterations. However, with our greedy method, once the parameters $\theta_t$ and the next iterates $x_k^{t+1}$ for $k \in \{1, \cdots, N\}$ have been calculated, $\theta_t$ is no longer required to be stored on the GPU, and can be saved to disk. Therefore GPU memory is constant with increasing training iterations for our greedy method. Suppose that training is terminated after iteration $T$, having learned the parameters $\theta_0, \cdots, \theta_T$. To minimise an unseen function $f$ with initial point $x_0$, we propose Algorithm 2.

---

**Algorithm 1** Training algorithm for greedy parameter learning in preconditioned gradient descent

---

1: **Input:** Functions $f_1, \cdots f_N$, initial points $x_1^0, \cdots, x_N^0$, final iteration $T$, regularization parameters $\lambda_0, \cdots, \lambda_T \geq 0$.
2: **for** $t = 0, 1, 2, \ldots, T$ **do**
3:      $\theta_t \in \arg\min_\theta g_{t,\lambda_t}(\theta)$
4:      **for** $k = 1, 2, \ldots, N$ **do**
5:          $x_k^{t+1} = x_k^t - G_{\theta_t} \nabla f_k(x_k^t)$
6:      **end for**
7: **end for**
8: **Output:** Learned parameters $\theta_0, \cdots, \theta_T$.

---

**Algorithm 2** Learned algorithm applied to a new function $f$

---

1: **Input:** Function $f$ with initial point $x_0$.
2: **for** $t = 0, 1, 2, \ldots$ **do**
3:      **if** $t \leq T$ **then**
4:          $x_{t+1} = x_t - G_{\theta_t} \nabla f(x_t)$
5:      **else**
6:          $x_{t+1} = x_t - G_{\theta_T} \nabla f(x_t)$
7:      **end if**
8: **end for**
9: **Output:** $x_{t+1}$.

---

## 4    CONVERGENCE RESULTS

This section contains convergence results for our learned Algorithm 2. Firstly, in Theorem 1 convergence is obtained on training functions as $T \to \infty$, without the need for the learned operators $G_{\theta_t}$ to have properties such as being symmetric or positive definite. Following this, in Theorem 2 we show convergence results with rates for a class of unseen functions if $\lambda_t$ is asymptotically non-vanishing. Before we present the convergence results, we require the following definitions, the first of which provides a condition for which the update rule (5) generalizes gradient descent (GD): $x_{t+1} = x_t - \alpha_t \nabla f(x_t)$ for $\alpha_t > 0$.

**Definition 1.** *We say that the family* $(G_\theta)$ *is GGD (generalizes gradient descent) if for all* $\alpha > 0$, *there exist parameters* $\theta$ *such that*

$$G_\theta = \alpha I. \tag{7}$$

Parametrizations that satisfy the GGD property are shown in section 5. Let $\tau = 1/L_{\text{train}}$, where $L_{\text{train}} = \max\{L_1, \cdots, L_N\}$ is the largest smoothness coefficient in the training data set. This choice of step size in gradient descent ensures convergence for all functions $f_k \in \mathcal{T}$. From this point forward, we assume $(G_\theta)$ is GGD, meaning there exists some $\tilde{\theta}$ such that $G_{\tilde{\theta}} = \tau I$. Furthermore, the GGD property can be leveraged to establish provable convergence for a set of unseen functions by introducing a penalty when the parameters deviate significantly from $\tilde{\theta}$. With this purpose, we define $R(\theta)$ in (6) as

$$R(\theta) := \frac{1}{2}\|\theta - \tilde{\theta}\|^2. \tag{8}$$

The next definition is to ensure the parametrized algorithm adopts the convergence properties of gradient descent on the training data.

**Definition 2.** *We say that* $\theta_t$ *is BGD (better than gradient descent) with regularization parameter* $\lambda_t$ *if*

$$g_{t,\lambda_t}(\theta_t) \leq g_{t,\lambda_t}(\tilde{\theta}) = g_{t,0}(\tilde{\theta}) = \frac{1}{N}\sum_{k=1}^{N} f_k\left(x_k^t - \tau\nabla f_k(x_k^t)\right). \tag{9}$$

In section 5 we introduce parameterizations $G_\theta$ for which the BGD property is easily obtained during training.

## 4.1 CONVERGENCE ON TRAINING DATA

**Theorem 1.** *Convergence on training data. Suppose that $\lambda_t \geq 0$ and $(\theta_t)_{t=0}^{\infty}$ is a BGD sequence of parameters. Then with Algorithm (1), we have $\nabla f_k(x_k^t) \to 0$ as $t \to \infty$ for all $k \in \{1, \cdots, N\}$.*

Note that in particular, this means that convergence in training is obtained even when $\lambda_t = 0$ for all $t$. Therefore, the learned preconditioners $G_t$ are never necessarily positive-definite. Convergence rates can also be obtained for training data, see Appendix Section 4.

## 4.2 CONVERGENCE ON UNSEEN DATA

We now show convergence on unseen data. Firstly, we show that if the regularization parameters $\lambda_t$ are eventually non-vanishing, then the learned parameters tend towards $\tilde{\theta}$.

**Lemma 1.** *If $\liminf_{t\to\infty} \lambda_t > 0$ and $(\theta_t)_{t=0}^{\infty}$ is BGD, then $\theta_t \to \tilde{\theta}$ as $t \to \infty$.*

This result is useful to ensure convergence on unseen data as if $G : \Theta \to \mathcal{L}(\mathcal{X})$ is continuous, then under the same conditions, $G_{\theta_t} \to G_{\tilde{\theta}} = \tau I$ as $t \to \infty$, i.e. our learned algorithm gets close to GD for large $t$. The idea is that we start with a method that fits the data very well leading to quick initial convergence, but in the interest of safety, over time we become closer to an algorithm with proved convergence, with $G_{\theta_t}$ positive-definite eventually.

**Theorem 2.** *Convergence on unseen data for regularized parameter learning*
*Assume that $G : \Theta \to \mathcal{L}(\mathcal{X})$ is continuous, $\theta_t$ is BGD and $\liminf_{t\to\infty} \lambda_t > 0$. Then, there exists a final training iteration $T$ such that for all $f \in \mathcal{F}_{L_{train}}$ and any starting point $x_0$, using Algorithm 2 (which depends on $T$), we have $\nabla f(x_t) \to 0$ as $t \to \infty$.*

Note that all training functions $f_k \in \mathcal{T}$ satisfies $f_k \in \mathcal{F}_{L_k} \subseteq \mathcal{F}_{L_{train}}$, and therefore Theorem 2 holds for all training functions. In practice, provable convergence can be verified during training. At iteration $T$, the regularization parameter $\lambda_T$ may be selected large enough such that $\|G_{\theta_T} - \tau I\| < \tau$, which guarantees convergence. A proof is provided with Proposition 6 in the Appendix. The following theorem presents the convergence rate obtained for test functions.

**Theorem 3.** *Convergence rates on unseen data for regularized parameter learning*
*Under the same assumptions as Theorem 2 and if $(x_t)_{t=1}^{\infty}$ is a bounded sequence then there exists a constant $C > 0$, such that*

$$f(x_t) - f(x^*) \leq \frac{C}{t}. \tag{10}$$

This result gives the worst-case convergence rate of the learned algorithm. In Section 6 we will see that the empirical performance of the learned algorithms may exceed that of NAG and L-BFGS.

## 5 LINEAR PARAMETRIZATIONS

In this section, we consider 'linear parametrizations' of $G$, defined below.

**Definition 3.** *We call $G$ a linear parameterization if $G : \Theta \to \mathcal{L}(\mathcal{X})$ is a linear map. This means there exists a linear operator $B_k^t \in \mathcal{L}(\Theta, \mathcal{X})$ such that*

$$G_\theta \nabla f_k(x_k^t) = B_k^t \theta. \tag{11}$$

The motivation is that when $G$ is a linear parametrization, each optimization problem (6) is convex (as it is the composition of a convex function with a linear function (Beck, 2014)). Therefore, learning comprises solving a sequence of convex optimization problems. In this case, there exist fast, provably convergent algorithms to find global solutions. Due to the speed of training enabled by linear parameterizations, we are able to learn algorithms up to significantly higher iterations. In Section 6, we see this enables algorithms to be learned up to iterations where a pre-selected tolerance has been satisfied. Four examples of linear parametrizations of $G$ are provided in Table 1. These parametrizations are used for the numerical experiments in Section 6. Due to the convexity of $g_{t,\lambda_t}$, the BGD property is easily verified during training for each parametrization.

**Lemma 2.** *All parametrizations $G_\theta$ in Table 1 satisfy the GGD property (7), and are all continuous with respect to their parameters.*

Table 1: Examples of linear parametrizations

| Label | Description | parametrization | # parameters |
|-------|-------------|-----------------|--------------|
| (PS) | Scalar step size | $G_{\alpha_t} = \alpha_t I, \alpha_t \in \mathbb{R}$ | 1 |
| (PP) | Pointwise operator | $G_{p_t} x = p_t \odot x, p_t, x \in \mathcal{X}$ | $\dim(\mathcal{X})$ |
| (PC) | Image convolution | $G_{\kappa_t} x = \kappa_t * x, \kappa_t \in \mathbb{R}^{m_1 \times m_2}$ | $m_1 m_2$ |
| (PF) | Full linear operator | $G_{P_t} = P_t \in \mathcal{L}(\mathcal{X})$ | $\dim(\mathcal{X})^2$ |

**Corollary 1.** *If the assumptions from Theorem 2 are satisfied, then for linear parametrizations in Table 1, we obtain the convergence results. Furthermore, if the sequence $(x_t)_{t=1}^{\infty}$ is bounded in Algorithm 2, we obtain the convergence rates as in Theorem 3.*

### 5.1 CLOSED-FORM SOLUTIONS

If each function $f_k \in \mathcal{T}$ can be written as a least-squares function, then the parameters $\theta_t$ at iteration $t$ have a closed-form solution.

**Proposition 3.** *For $k \in \{1, \cdots, N\}$, let $f_k : \mathcal{X} \to \mathbb{R}$ be given by $f_k(x) = \frac{1}{2}\|A_k x - y_k\|^2$, with corresponding $y_k \in \mathcal{Y}$, for a Hilbert space $\mathcal{Y}$, and linear operator $A_k \in \mathcal{L}(\mathcal{X}, \mathcal{Y})$. For a linear parametrization $G$, let $B_k^t$ be given as in (11). Then $\theta_t$ given by*

$$\theta_t = \left( \lambda_t I_\Theta + \frac{1}{N} \sum_{k=1}^{N} (A_k B_k^t)^*(A_k B_k^t) \right)^\dagger \left( \lambda_t \tilde{\theta} + \frac{1}{N} \sum_{k=1}^{N} (B_k^t)^* \nabla f_k(x_k^t) \right) \tag{12}$$

*is a solution to (6), where $M^\dagger$ represents the Moore–Penrose pseudoinverse of a linear operator $M$.*

Note that we recover the closed-form equation for exact line search for a scalar step size (Nocedal & Wright, 2006a) with $\lambda_t = 0, N = 1$ for the parametrization (PS) in Table 1. Therefore the optimization problem (6) can be seen as an extension of exact line search to include linear operators. Calculations for the closed-form solutions for the parametrizations in Table 1 are detailed in Appendix Section 5. In general, we require optimization algorithms to approximate $\theta_t$. Due to the optimization problem being convex, we provide gradient calculations and smoothness constants for these parametrizations in Appendix Section E.1. Therefore, we do not require step size tuning for learning parameters $\theta_t$.

## 6 NUMERICAL EXPERIMENTS

**The optimization problem.** In this section, we test the four linear parametrizations in Table 1 on two inverse problems in imaging: image deblurring and CT. We consider linear inverse problems, defined by receiving an observation $y \in \mathcal{Y}$, generated from a ground-truth $x_{\text{true}}$ via some linear forward operator $A : \mathcal{X} \to \mathcal{Y}$, such that $y = Ax_{\text{true}} + \varepsilon$, where $\varepsilon \in \mathcal{Y}$ is some random noise, and the goal is to recover $x_{\text{true}}$. In this case, we create observations from given ground-truth data as described above. Once these observations have been created, the ground-truth data are no longer used. For both experiments, $\mathcal{X} = \mathbb{R}^{h_1 \times h_2}, \mathcal{Y} = \mathbb{R}^{h_3 \times h_4}$ for $h_1, h_2, h_3, h_4 \in \mathbb{N}$, and $\varepsilon$ is noise sampled from a zero-mean Gaussian distribution. To approximate $x_{\text{true}}$ from $y$, we solve

$$\min_x \left\{ f(x) := \frac{1}{2}\|Ax - y\|^2 + \alpha H_\epsilon(x) \right\}, \tag{13}$$

for a fixed regularization parameter $\alpha$. The regularizer $H_\epsilon$ is the Huber Total Variation (Rudin et al., 1992; Huber, 1992) defined by

$$H_\epsilon(x) = \sum_{i,j=1}^{h_1, h_2} h_\epsilon \left( \sqrt{(\mathrm{D}x)_{i,j,1}^2 + (\mathrm{D}x)_{i,j,2}^2} \right), \quad h_\varepsilon(s) = \begin{cases} \frac{1}{2\epsilon}s^2, & \text{if } |s| \leq \epsilon \\ |s| - \frac{\epsilon}{2}, & \text{otherwise,} \end{cases} \tag{14}$$

where finite difference operator $\mathrm{D} : \mathbb{R}^{h_1 \times h_2} \to \mathbb{R}^{h_1 \times h_2 \times 2}$ is defined in Chambolle & Pock (2016). Note that this choice of regularizer makes the function $f$ non-quadratic. We take $\epsilon = 0.01$ and

normalize the forward operator in both cases so that $\|A\| = 1$. Then, each function $f$ is $L$-smooth, where $L = 1 + \frac{8\alpha}{\epsilon}$ (Chambolle & Pock, 2016).

**Learning parameters.** For each parametrization in Table 1, to learn parameters $\theta_t$ we apply NAG for solving the optimization problem 6. We initialize as $\theta_t^0 = \tilde{\theta}$ for $t = 0$, and $\theta_t^0 = \theta_{t-1}$ for $t > 0$. NAG is stopped when $\|\nabla g_{t,\lambda_t}(\theta_t^\ell)\|/\|\nabla g_{t,\lambda_t}(\theta_t^0)\| < 10^{-3}$, or when $\ell = \ell_{\text{stop}} = 5000$. For both problems, we use a training set of 100 functions for parametrizations (PS), (PP), and (PC). For (PF), the model is trained using 1000 functions and is only implemented for the small-scale CT problem. Testing is performed on a separate set of 100 functions for all parametrizations. The learned convolutional kernels (PC) have dimensions $h_1 \times h_2$, matching the size of the images in $\mathcal{X}$.

**Evaluation.** Given a dataset of functions $f_1, \ldots, f_N$, the mean value at iteration $t$ is defined as $F(x_t) = \frac{1}{N}\sum_{k=1}^N f_k(x_k^t)$. Furthermore, we define "function optimality" for a function $f$ with minimizer $x_f^*$ at iteration $t$ by $(f(x_t) - f(x_f^*))/(f(x_0) - f(x_f^*))$. For a function $f$, its approximate minimizer $x_f^* \in \mathcal{X}$ is calculated using NAG. For a dataset of functions, we visualize the maximum and minimum function optimality over the dataset and the function optimality for $F$. The learned algorithms are compared to NAG with backtracking (Beck & Teboulle, 2009) and L-BFGS with the Wolfe conditions (Wolfe, 1969). Computations were performed on an Nvidia RTX 3600 12GB GPU.

## 6.1 Image deblurring

**Problem details.** The forward operator $A$ in (13) is a Gaussian blur with a $5 \times 5$ kernel size and a standard deviation $\sigma = 1.5$. We use the STL-10 dataset (Coates et al., 2011) with greyscale images of size $96 \times 96$ as $\mathcal{X}$. The noise $\varepsilon$ is modeled with a standard deviation of $2.5 \times 10^{-3}$, and we set $\alpha = 10^{-5}$, resulting in $L = 1.008$. The initial point $x_0 = y \in \mathcal{Y} = \mathcal{X}$ is chosen as the observation.

**Training details.** Training with the greedy method was performed up to iteration $T = 250$ with $\lambda_t = 0$ for all $t$ for the parametrizations (PS), (PP), (PC). This means 250 parameters were learned for (PS) and 2304000 for both (PP) and (PC). The total training time for (PS) was 2.8 minutes, 17 minutes for (PP) and 9.2 hours for (PC).

**Visualising learned preconditioners.** Figure 1a shows that the learned scalar parameters (PS) eventually fluctuate around $2/L$, which is outside of the range of provable convergence of gradient descent with a constant step size. Despite this, the learned algorithm leads to convergence on training data as $t \to \infty$ by Theorem 1. In Figure 1b, we also see negative values for the pointwise parametrization (PP). The learned convolutional kernels (PC) in Figure 1c also contain positive and negative values and are predominantly weighted towards the center, suggesting that information from neighboring pixels is prioritized over more distant ones. As the number of iterations increases, the kernels exhibit increasing similarity, though no formal convergence result for $\theta_t$ has been established when $\lambda_t = 0$.

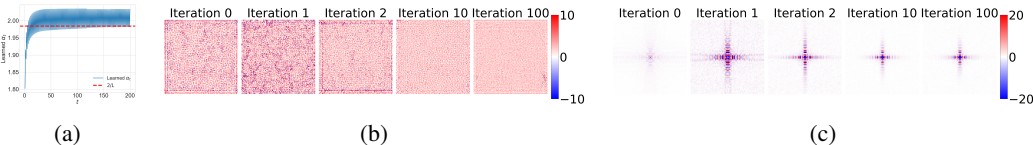

(a)           (b)           (c)

Figure 1: Learned parameters for the image deblurring problem with $\lambda_t = 0$ for all $t \in \{0, 1, \cdots, T\}$. (a) Learned scalar parameters (PS) for $t \in \{0, 1, \cdots, 200\}$ compared to $2/L$. (b) Learned $96 \times 96$ pointwise operators (PP) restricted to the interval $[-10, 10]$, against iteration $t$ for $t \in \{0, 1, 2, 10, 100\}$. (c) Learned $96 \times 96$ convolutional kernels (PC) restricted to the interval $[-20, 20]$, against iteration $t$ for $t \in \{0, 1, 2, 10, 100\}$.

**Learned algorithm performance.** Figure 2a shows that the learned parametrizations (PS), (PP), and (PC) generalize well to unseen data due to the closeness of the train and test curves. (PP) performs comparably to (PS) for this example, despite having an equal number of parameters as (PC), which captures global information of the image, rather than only pixel-level details. Note that (PC) reaches the tolerance of $10^{-7}$ before training completes, as our method allows us to learn an algorithm that runs for sufficient iterations to meet a pre-specified tolerance. Figure 2b shows (PC) significantly outperforms both NAG and L-BFGS on the test data, reaching a tolerance of $10^{-7}$ in just over 100

iterations on average, compared with about 600 for L-BFGS and NAG. We also see that the worst-case performance of (PC) outperforms the best-case performance of NAG and L-BFGS. Figure 2c shows (PC) also outperforms other algorithms when considering wall-clock time. Appendix Section F.1 explores the impact of different kernel sizes on the performance of (PC), and Appendix Section F.5 shows the number of iterations to reach a specified tolerance for different algorithms.

**Comparison to a hand-crafted convolutional preconditioner.** In Figures 2b and 2c we also evaluate a hand-crafted convolutional algorithm. In particular, we consider the preconditioner $(\delta I + A^*A)^{-1}$ for $\delta = 0.2$, which corresponds to a convolution with the kernel shown in Appendix Section F.2. We evaluate the update rule given by $x_{t+1} = x_t - \gamma_t(\delta I + A^*A)^{-1}\nabla f(x_t)$ for a function $f$ and a scalar step $\gamma_t$ found using backtracking line search, and denote this algorithm PGD. Figures 2b and 2c show that the learned convolutional algorithm significantly outperforms this hand-crafted algorithm.

**Regularization.** We use regularization at iteration $T$ for the parametrizations (PS) and (PP) to ensure convergence when applying Algorithm 2 to further iterations. The (PC) parametrization was not considered as it has already reached a suitable tolerance within the training iterations. As discussed in Section 4, we may select $\lambda_T$ large enough to guarantee convergence. We find $\lambda_T = 4.012 \times 10^{-7}$ and $\lambda_T = 1.953 \times 10^{-9}$ guarantee convergence for (PS) and (PP), respectively. Figure 2d shows that the learned algorithm diverges when learned without regularization, but converges if the regularization parameter $\lambda_T$ is chosen large enough.

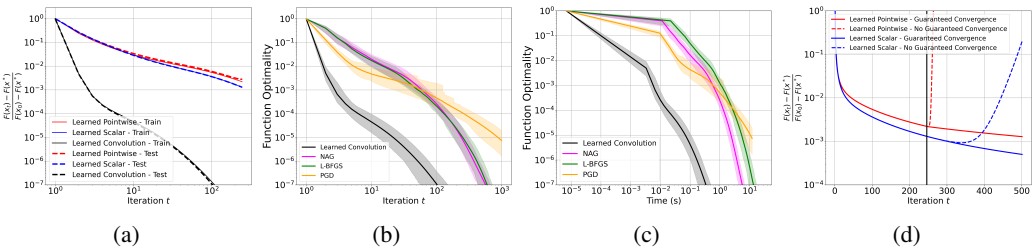

(a)        (b)        (c)        (d)

Figure 2: (a) Performance of the learned methods on training versus test data within training iterations. (b) Test performance versus benchmark optimization algorithms within training iterations. Intervals around each mean represent maximum and minimum values over the dataset. (c) Comparison with Wall-clock time on test data. (d) Performance on training data beyond training iterations for the (PP) and (PS) learned with and without guaranteed convergence. The vertical black line indicates the final training iteration $T$.

**Reconstruction comparison.** Figure 3 demonstrates that the learned convolutional algorithm achieves high-quality image reconstruction in 10 iterations, whereas NAG produces lower-quality reconstructions at the same point.

**Greedy learning vs unrolling.** We now compare the time taken for training with the greedy learning approach versus unrolling. For unrolling, we fix $T = 10$ iterations and jointly learn the parameters $\theta_0, \cdots, \theta_T$ (all initialized as $\tilde{\theta}$) in the update rule (5) with the (PC) parametrization. The same training dataset as the greedy method is used with a batch size of 4 and the loss function defined in equation (4). Parameters are learned using Adam (Kingma, 2014), with the learning rate selected via grid search. The unrolling method was trained for 29900 epochs, taking approximately 27 hours. Figure 4 shows similar performance, although the greedy approach took considerably less time with only 22 minutes to learn parameters.

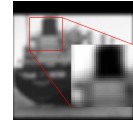 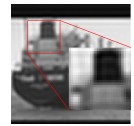 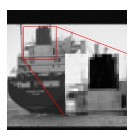 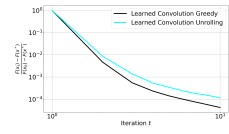
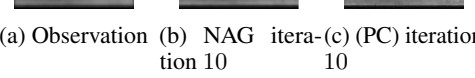

(a) Observation (b) NAG itera-(c) (PC) iteration
                    tion 10       10

Figure 3: A Comparison of reconstructions for the deblurring problem.

Figure 4: Performance of the learned unrolled algorithm versus the greedy learned algorithm on test data.

## 6.2 COMPUTED TOMOGRAPHY

Now the forward operator $A$ in (13) is the Radon transform in 2D and we simulate CT measurements using ODL (Adler et al., 2017) with a parallel-beam geometry and projection angles evenly distributed over a 180-degree range. For the dataset, we use ground-truth images in the SARS-CoV-2 CT-scan dataset (Soares et al., 2020), and in optimization take the initial point $x_0 = 0 \in \mathcal{X}$.

### 6.2.1 LARGE-SCALE CT PROBLEM

**Problem details.** We use 360 projection angles and take $\mathcal{X} = \mathbb{R}^{256 \times 256}$ and $\mathcal{Y} = \mathbb{R}^{360 \times 360}$. The noise $\varepsilon$ is modeled with standard deviation $10^{-3}$, and we set $\alpha = 10^{-6}$, resulting in $L = 1.0008$.

**Training details.** Greedy training was performed up to iteration $T = 200$ with $\lambda_t = 0$ for all iterations $t$ for the (PS), (PP), and (PC) parametrizations. The total time for training (PS) was about 33 minutes, for (PP) was about 10 hours, and (PC) took approximately 53 hours.

**Visualising learned preconditioners.** Figure 5a shows that the learned convolutional kernels for the CT problem contain both positive and negative values, and are predominantly weighted toward the center of the kernel. Figure 5b shows that the learned pointwise operators look similar to images in the SARS-CoV-2 dataset, and exhibit oscillations between consecutive iterations, with many values falling outside the interval $(0, 2/L)$. Likewise, Figure 5c shows that the learned scalar values again fluctuate above and below $2/L$, similar to the behavior observed for the deblurring problem.

**Learned algorithm performance.** Figures 5f and 5g show that the learned convolutional algorithm achieves a good reconstruction faster than NAG. Furthermore, Figure 6a shows that the learned parametrizations (PS), (PP), and (PC) generalize well to unseen data for the CT problem. Similar to the deblurring problem, Figure 6b shows that the learned (PC) parametrization outperforms NAG and L-BFGS on the CT test data, reaching a tolerance of $10^{-8}$ in an average of approximately 90 iterations, compared with over 150 for both L-BFGS and NAG. However, we see that the worst-case performance of (PC) does not beat the best-case performance of NAG and L-BFGS. However, Figure 6c shows that the worst-case wall-clock time for the learned convolutional algorithm to reach a tolerance of $10^{-8}$ is less than the best-case wall-clock time for NAG.

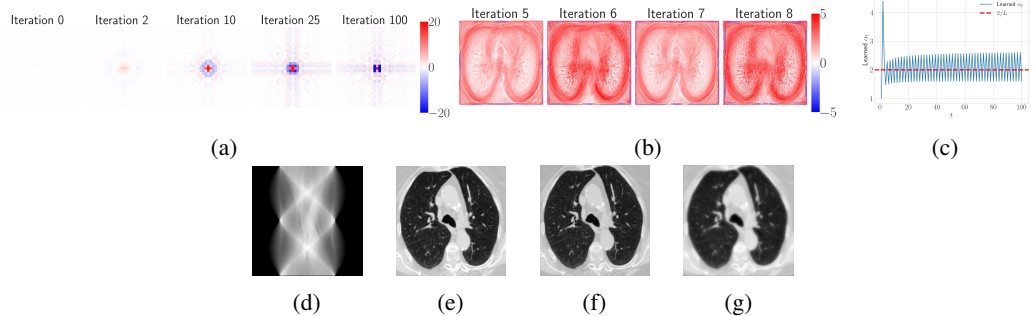

Figure 5: (a) Learned kernels restricted to the interval $[-20, 20]$ for $t \in \{0, 2, 10, 25, 100\}$, cropped to the center $32 \times 32$. Uncropped images are shown in Appendix Section F.4. (b) Learned pointwise operators for $t \in \{5, 6, 7, 8\}$ restricted to $[-5, 5]$. Extended iterations are shown in Appendix Section F.4. (c) Learned scalars for $t \in \{0, 1, \cdots, 100\}$. (d) Example CT observation. (e) Reconstruction by minimizing (13). (f) (PC) reconstruction at iteration 20. (g) NAG reconstruction at iteration 20.

### 6.2.2 SMALL-SCALE CT PROBLEM

**Problem details.** We use 90 projection angles and extract $40 \times 40$ pixel crops from the center of each ground-truth image in the dataset. The noise $\varepsilon$ is modeled with a standard deviation of $10^{-2}$, and we set $\alpha = 10^{-4}$, resulting in $L = 1.08$.

**Training details.** Greedy training was performed up to iteration $T = 200$ with $\lambda_t = 0$ for all iterations $t$ for the (PS), (PP), and (PC) parametrizations. The total time for training (PS) was about 10 minutes, for (PP) was about 67 minutes, and (PC) took approximately 10 hours. For the (PF)

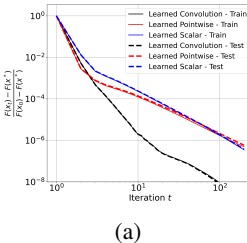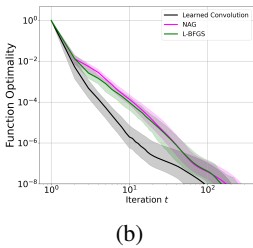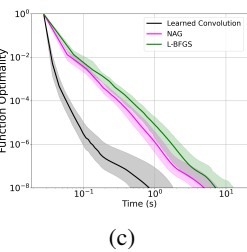

(a)        (b)        (c)

Figure 6: Performance of learned algorithms for the large-scale CT problem. (a) Train versus test set performance of the learned parameterizations. (b) Test performance versus benchmark optimization algorithms. (c) Wall-clock time test performance versus benchmark optimization algorithms.

parametrization, training was performed up to iteration $T = 11$ with $\lambda_t = 0$ for all $t$. Furthermore, the (PF) parametrization was trained with regularization such that $\lambda_t = 10^{-10}$ for $t < T = 101$ iterations. At iteration $101$, the learned operator $G_{\theta_T}$ satisfied $\|G_{\theta_T} - \tau I\| < \tau$, guaranteeing convergence on iterations $t \geq T$. For each iteration $t$, solving the optimization problem (6) with the (PF) parametrization took one hour.

**Learned algorithm performance.** Figure 7a shows that the learned parametrizations (PS), (PP), and (PC) generalize well to unseen data for the CT problem. Again the learned (PC) parametrization outperforms NAG and L-BFGS on the CT test data as shown by Figure 7b, reaching a tolerance of $10^{-10}$ in approximately 30 iterations, compared with about 80 for L-BFGS and NAG. Figure 7c shows that (PC) also outperforms in terms of wall-clock time. Learned preconditioners and reconstruction comparisons can be found in Appendix Section F.3.

**Full operators.** The full parametrization (PF) shows signs of overfitting, as it does not generalize well to test data. It performs well in the first two iterations, but then diverges. The (PF) parametrization with regularization mitigates this issue, as the generalization performance is seen to improve. Figure 7b shows it initially converges quickly but its speed decreases later due to regularization. This is because, with increasing iterations, the learned update gets closer to gradient descent.

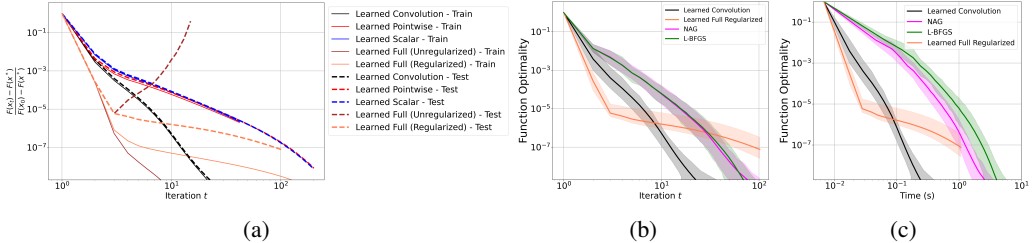

(a)        (b)        (c)

Figure 7: Performance of learned algorithms for the small-scale CT problem. (a) Train versus test set performance of the learned parameterizations. (b) Test performance versus benchmark optimization algorithms. (c) Wall-clock test performance versus benchmark optimization algorithms.

## 7 CONCLUSIONS

Our contribution is a novel L2O approach for minimizing unconstrained convex problems with differentiable objective functions. Our method employs a greedy strategy to learn a linear operator at each iteration of an optimization algorithm, meaning that GPU memory requirements are constant with the number of training iterations. Parameter learning in our framework corresponds to solving convex optimization problems, enabling the use of fast algorithms. Both factors allow training over a large number of training iterations, which would otherwise be prohibitively expensive. Furthermore, we obtain convergence results on the training set even when the preconditioner is neither symmetric nor positive definite, and for a class of unseen functions under certain conditions. The numerical results on imaging inverse problems demonstrate that our approach with a novel convolutional parametrization outperforms NAG and L-BFGS.

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

## A  LIMITATIONS

The main limitation of our method lies in the fact that we learn

## B  FURTHER NOTATION

The following notation is required in this section.

A function $f : \mathcal{X} \to \mathbb{R}$ is strongly convex with parameter $\mu > 0$ if $f - \frac{\mu}{2} \| \cdot \|^2$ is convex. We say $f \in \mathcal{F}_{L,\mu}$ if $f \in \mathcal{F}_L$ and $f$ is $\mu$-strongly convex.

If $\dim(\mathcal{X}) = n$ and $\dim(\mathcal{Y}) = m$, denote an orthonormal basis of $\mathcal{X}$ by $\{e_1, \cdots, e_n\}$, and an orthonormal basis of $\mathcal{Y}$ by $\{\tilde{e}_1, \cdots, \tilde{e}_n\}$, then $A$ can be uniquely determined by $mn$ scalars $\gamma_{ij}$ for $i \in \{1, \cdots, m\}, j \in \{1, \cdots, n\}$: $A(e_i) = \sum_{j=1}^n \gamma_{ij} \tilde{e}_j$, and denote $[A]_{ij} = \gamma_{ij}$. For $x, y \in \mathcal{X}$, define the pointwise product $x \odot y$ by

$$[x \odot y]_j = [x]_j [y]_j. \tag{15}$$

. Denote $\mathbf{1} \in \mathcal{X}$ to be such that $[\mathbf{1}]_j = 1$ for $j \in \{1, \cdots, n\}$. For operators $A, B \in \mathcal{L}(\mathcal{X})$, and elements $x, y, z \in \mathcal{X}$, define the linear operators $A \odot B$ and $x \otimes y$ by

$$[A \odot B]_{ij} := [A]_{ij} [B]_{ij}, \tag{16}$$

$$(x \otimes y)z := \langle y, z \rangle x, \tag{17}$$

with the property that

$$[x \otimes y]_{qi} = \langle y, e_i \rangle \langle x, e_q \rangle = [x]_q [y]_i. \tag{18}$$

For two linear operators $A, B \in \mathcal{L}(\mathcal{X})$, define $A \otimes B$ by

$$[A \otimes B]_{ij,kl} = [A]_{ik} [B]_{jl}. \tag{19}$$

For a linear operator $A \in \mathcal{L}(\mathcal{X})$ and $x \in \mathcal{X}$,

$$[Ax]_i = \sum_{j=1}^n [A]_{ij} [x]_j. \tag{20}$$

## C  PROOFS FOR SECTION 3

**Proposition 4.** *Assume that $f : \mathcal{X} \to \mathbb{R}$ is convex, continuously differentiable, and has a global minimum. Then for a point $z \in \mathcal{X}$ if there exists some $x^* \in \arg\min_x f(x)$ such that $[z]_i = [x^*]_i$, then $[\nabla f(z)]_i = 0$.*

*Proof.* Let $g : \mathbb{R} \to \mathbb{R}$ be defined by $g(t) := f(z + te_i)$, then $g$ is convex as for $\alpha \in [0, 1], t_1, t_2 \in \mathbb{R}$, we have $g(\alpha t_1 + (1 - \alpha)t_2) = f(\alpha(z + t_1 e_i) + (1 - \alpha)(z + t_2 e_i)) \leq \alpha g(t_1) + (1 - \alpha)g(t_2)$. Note that $g'(0) = [\nabla f(z)]_i$. Assume there exists $\delta \neq 0$ such that $g(\delta) < g(0)$, then $g(\delta) < g(0) = f(z)$, which is a contradiction of $x^*$ being an optimal point, as one can take $z = x^*$. Therefore $g$ achieves a minimum at $t = 0$, then $[\nabla f(z)]_i = 0$. $\qquad\square$

*Proof of Proposition 1.* Choose the vector $p \in \mathcal{X}$ such that

$$[p]_i = \begin{cases} \frac{[x_0 - x_*]_i}{[\nabla f(x_0)]_i}, & \text{if } [\nabla f(x_0)]_i \neq 0, \\ 0, & \text{otherwise,} \end{cases} \tag{21}$$

and let $I = \{i : [\nabla f(x_0)]_i \neq 0\}$ Then for any $i \in I$, we have

$$[x_0 - p \odot \nabla f(x_0)]_i = [x_0]_i - [p]_i [\nabla f(x_0)]_i$$

$$= [x_0]_i - \frac{[x_0 - x^*]_i}{[\nabla f_k(x_0)]_i} [\nabla f(x_0)]_i$$

$$= [x^*]_i.$$

Thus, by proposition 4, $[\nabla f(x_0 - p \odot \nabla f(x_0))]_i = 0$, for all $i \in I$, and similarly $[\nabla f(x_0 - p \odot \nabla f(x_0))]_i = 0$, for all $i \notin I$, and therefore $\nabla f(x_0 - p \odot \nabla f(x_0)) = 0$, meaning that $x_0 - p \odot \nabla f(x_0) \in \arg\min_x f(x)$ as required. $\qquad\square$

***Proof of Proposition 2.*** We require

$$
\begin{cases}
x_1^* & = x_1^0 - P\nabla f_1(x_1^0), \\
& \vdots \\
x_N^* & = x_N^0 - P\nabla f_N(x_N^0).
\end{cases}
$$

Each of these equations gives $n$ linear equations in $n^2$ unknowns. There are $N$ such equations and so we have $nN$ linear equations in $n^2$ unknowns. Rewritten, these read

$$
P\left[\nabla f_1(x_1^0)|\cdots|\nabla f_N(x_N^0)\right] = \left[x_1^0 - x_1^*|\cdots|x_N^0 - x_N^*\right]. \tag{22}
$$

Such a $P$ exists if $nN \leq n^2$, which is equivalent to $N \leq n$, and if the columns of $\left[\nabla f_1(x_1^0)|\cdots|\nabla f_N(x_N^0)\right]$ are linearly independent. $\qquad\square$

# D    PROOFS FOR SECTION 4

The following lemma is required to prove the convergence of our learned method.

**Lemma 3.** *Define $F : \mathcal{X}^N \to \mathbb{R}$ by*

$$
F(x) = \frac{1}{N}\sum_{k=1}^{N} f_k(x_k), \quad x = (x_1, x_2, \ldots, x_N) \in \mathcal{X}^N. \tag{23}
$$

*Then*

  *1. Each $f_k \in \mathcal{F}_{L_k}$ implies $F \in \mathcal{F}_{L_F}$, with $L_F = \frac{1}{N}L_{train}$ where $L_{train} = \max\{L_1, \cdots, L_N\}$.*

  *2. Each $f_k \in \mathcal{F}_{L_k,\mu_k}$ implies $F \in \mathcal{F}_{L,\mu_F}$ with $\mu_F = \frac{1}{N}\mu_{min}$ where $\mu_{min} = \min\{\mu_1, \cdots, \mu_N\}$.*

*Proof.* We have

$$
\nabla F(x) = \frac{1}{N}\left(\nabla f_1(x_1), \cdots, \nabla f_N(x_N)\right), \tag{24}
$$

and for any $y \in \mathcal{X}^N$,

$$
\|x - y\| = \sqrt{\sum_{k=1}^{N}\|x_k - y_k\|^2}.
$$

Then

$$
\|\nabla F(x) - \nabla F(y)\| = \frac{1}{N}\sqrt{\sum_{k=1}^{N}\|\nabla f_k(x_k) - \nabla f_k(y_k)\|^2}
$$

$$
\leq \frac{1}{N}\sqrt{\sum_{k=1}^{N} L_k^2\|x_k - y_k\|^2} \quad (L_k\text{-smoothness of } f_k.)
$$

$$
\leq \frac{\max\{L_1, \cdots, L_N\}}{N}\|x - y\|,
$$

which proves 1.

For strong convexity, it is required to show that $x \mapsto (F(x) - \frac{\min\{\mu_1,\cdots,\mu_N\}}{N}\|x\|^2)$ is convex. We have

$$
F(x) - \frac{\min\{\mu_1, \cdots, \mu_N\}}{N}\|x\|^2 = \frac{1}{N}\sum_{k=1}^{N}(f_k(x_k) - \min\{\mu_1, \cdots, \mu_N\}\|x_k\|^2). \tag{25}
$$

Notice that due to the strong convexity of $f_k$ for all $k$, and that $\mu_k \geq \min\{\mu_1, \cdots, \mu_N\}$,

$$
x_k \mapsto (f_k(x_k) - \min\{\mu_1, \cdots, \mu_N\}\|x_k\|^2) \tag{26}
$$

is convex. Therefore the function $x \mapsto (F(x) - \frac{\min\{\mu_1,\cdots,\mu_N\}}{N}\|x\|^2)$ is convex as it is the sum of convex functions, as required. $\qquad\square$

**Theorem 1. Convergence on training data.**
Suppose $\lambda_t \geq 0$. If $\theta_t$ is $BGD$ then with Algorithm (1), we have

$$\nabla f_k(x_k^t) \to 0 \text{ as } t \to \infty, \tag{27}$$

for all $k \in \{1, \cdots, N\}$.

**Bonus: Convergence rates**
Furthermore, if we denote

$$x_0 = (x_1^0, \cdots, x_N^0), \quad x^* = (x_1^*, \cdots, x_N^*), \tag{28}$$

then

$$F(x_t) - F(x^*) \leq \frac{\max\{L_1, \cdots, L_N\}}{2tN} \|x_0 - x^*\|^2. \tag{29}$$

If, in addition, each $f_k$ is $\mu_k$-strongly convex, then we have linear convergence given by

$$F(x_t) - F(x^*) \leq \left(1 - \frac{\max\{L_1, \cdots, L_N\}}{\min\{\mu_1, \cdots, \mu_N\}}\right)^t (F(x_0) - F(x^*)). \tag{30}$$

Note that this result gives a worst-case convergence bound among train functions. However, provable convergence is still acquired. Also, note that this is not an issue for a function class with constant smoothness and strongly convex parameters.

*Proof.* As $\theta_t$ is $BGD$, we have that

$$F(x_{t+1}) = g_{t,\lambda_t}(\theta_t) \leq g_{t,\lambda_t}(\tilde{\theta})$$

$$= \frac{1}{N} \sum_{k=1}^{N} f_k \left(x_k^t - \tau \nabla f_k(x_k^t)\right)$$

$$= F\left(x_t - \tau \left(\nabla f_1(x_1^t), \cdots, \nabla f_N(x_N^t)\right)\right)$$

$$= F\left(x_t - \tau N \nabla F(x_t)\right)$$

$$= F\left(x_t - \tau_F \nabla F(x_t)\right),$$

where $\tau_F = \frac{1}{L_F}$.

$F$ is $L_F$-smooth as each $f_k$ is $L_k$-smooth and $\mu$-strongly convex if each $f_k$ is $\mu_k$-strongly convex, where

$$L_F = \frac{\max\{L_1, \cdots, L_N\}}{N}$$

$$\mu_F = \frac{\min\{\mu_1, \cdots, \mu_N\}}{N}.$$

Using standard properties of $L$-smoothness and $\mu$-strong convexity we have that

$$F(x_{t+1}) \leq F(x_t) - \frac{1}{2L_F} \|\nabla F(x_t)\|^2, \tag{31}$$

$$\|\nabla F(x_t)\|^2 \geq 2\mu_F(F(x_{t+1}) - F(x^*)), \text{ if } F \text{ is } \mu_F\text{-strongly convex} \tag{32}$$

and therefore, using standard convergence rate results of gradient descent (Nesterov et al., 2018), we have

$$F(x_t) - F(x^*) \leq \frac{L_F}{2t} \|x_0 - x^*\|^2, \tag{33}$$

as $F$ is $L_F$-smooth. If $F$ is also $\mu_F$-strongly convex we have

$$F(x_t) - F(x^*) \leq \left(1 - \frac{L_F}{\mu_F}\right)^t (F(x_0) - F(x^*)). \tag{34}$$

In both cases, we have that $\|\nabla F(x_t)\|^2 = \frac{1}{N^2} \sum_{k=1}^{N} \|\nabla f_k(x_k^t)\|^2 \to 0$ as $t \to \infty$, which implies that $\nabla f_k(x_k^t) \to 0$ as $t \to \infty$ for all $k \in \{1, \cdots, N\}$. $\qquad \square$

We have proved convergence for the mean of our train functions. The following proposition proves the same convergence rate holds for each function in our training set.

**Proposition 5.** *Suppose we have a convergence rate for F of*

$$F(x_t) - F^* \leq C\rho(t), \tag{35}$$

*for some constant $C > 0$. Then the convergence rate for all $f_k \in \mathcal{T}$ is given by*

$$f_k(x_k^t) - f_k^* \leq c\rho(t), \tag{36}$$

*for some constant $c > 0$.*

*Proof.* Let $k \in \{1, \cdots, N\}$. Note that by the definition of $F$, we have that

$$f_k(x_k^t) - f_k^* \leq \sum_{i=1}^{N} f_i(x_i^t) - f_i^* \tag{37}$$

$$= N(F(x_t) - F^*) \tag{38}$$

$$\leq NC\rho(t) \tag{39}$$

$$= c\rho(t), \tag{40}$$

for $c = NC$.

$\square$

### D.1 UNSEEN DATA

***Proof of Lemma 1.*** Firstly,

$$g_{t,\lambda_t}(\tilde{\theta}) = \frac{1}{N} \sum_{k=1}^{N} f_k\left(x_k^t - \tau \nabla f_k(x_k^t)\right) \to \frac{1}{N} \sum_{k=1}^{N} f_k^* \text{ as } t \to \infty. \tag{41}$$

Note also that as $(\theta_t)_{t=0}^{\infty}$ is a $BGD$ sequence of parameters then $\frac{1}{N} \sum_{k=1}^{N} f_k^* \leq g_{t,\lambda_t}(\theta_t) \leq g_{t,\lambda_t}(\tilde{\theta})$ and so $g_{t,\lambda_t}(\theta_t) \to \frac{1}{N} \sum_{k=1}^{N} f_k^*$ as $t \to \infty$ as $g_{t,\lambda_t}(\theta_t) \geq \frac{1}{N} \sum_{k=1}^{N} f_k^*$. Furthermore,

$$g_{t,\lambda_t}(\tilde{\theta}) - g_{t,\lambda_t}(\theta_t) = -\frac{\lambda_t}{2}\|\theta_t - \tilde{\theta}\|^2 + \frac{1}{N} \sum_{k=1}^{N} f_k\left(x_k^t - \tau \nabla f_k(x_k^t)\right)$$

$$- \frac{1}{N} \sum_{k=1}^{N} f_k(x_k^t - G_{\theta_t} \nabla f_k(x_k^t))).$$

Therefore,

$$0 = \lim_{t \to \infty} g_{t,\lambda_t}(\tilde{\theta}) - g_{t,\lambda_t}(\theta_t) = \lim_{t \to \infty} -\frac{\lambda_t}{2}\|\theta_t - \tilde{\theta}\|^2. \tag{42}$$

Now, $\liminf_{t \to \infty} \lambda_t > 0$ implies that

$$\frac{\lambda}{2}\|\theta_t - \tilde{\theta}\|^2 \to 0 \text{ as } t \to \infty. \tag{43}$$

In particular, $\theta_t \to \tilde{\theta}$ as $t \to \infty$, as required. $\square$

**Lemma 4.** *Suppose $\liminf_{t \to \infty} \lambda_t > 0$, $G : \Theta \to \mathcal{L}(\mathcal{X})$ is continuous and at each training iteration $\theta_t$ is BGT. Then for any $\nu$ such that $0 \leq \nu < \tau$, there exists an iteration $T$ such that*

$$\|G_{\theta_T} - \tau I\| \leq \nu < \tau. \tag{44}$$

*Proof.* By Lemma 1, $\theta_t \to \tilde{\theta}$ as $t \to \infty$, therefore as $G_\theta$ is continuous in $\theta$ we have $G_{\theta_t} \to \tau I$ as $t \to \infty$. Therefore for any $\nu > 0$ there exists some iteration $T > 0$ such that

$$\|G_{\theta_t} - \tau I\| \leq \nu \tag{45}$$

for all $t \geq T$, in particular for $t = T$.

$\square$

***Proof of Theorem 2.*** By Lemma 4, for any tolerance $\nu < \tau$ there exists an iteration $T$ such that

$$\|G_{\theta_T} - \tau I\| \leq \nu. \tag{46}$$

Let

$$G_{\theta_T} = \tau I + M, \tag{47}$$

then

$$\|M\| \leq \nu. \tag{48}$$

Using $L$-smoothness of $f$, we have

$$
\begin{aligned}
f(x - G_{\theta_T} \nabla f(x)) &\leq f(x) - \langle G_{\theta_T} \nabla f(x), \nabla f(x) \rangle + \frac{L}{2} \|G_{\theta_T} \nabla f(x)\|^2 \\
&= f(x) - \left\langle G_{\theta_T} \nabla f(x), \nabla f(x) - \frac{L}{2} G_{\theta_T} \nabla f(x) \right\rangle \\
&= f(x) - \left\langle (\tau I + M) \nabla f(x), \nabla f(x) - \frac{L}{2} (\tau I + M) \nabla f(x) \right\rangle \\
&= f(x) - \left\langle \tau \nabla f(x) + M \nabla f(x), \nabla f(x) - \frac{L}{2} \tau \nabla f(x) - \frac{L}{2} M \nabla f(x) \right\rangle \\
&= f(x) - \left\langle \tau \nabla f(x) + M \nabla f(x), \left(1 - \frac{L\tau}{2}\right) \nabla f(x) - \frac{L}{2} M \nabla f(x) \right\rangle \\
&= f(x) - \tau \left(1 - \frac{L\tau}{2}\right) \|\nabla f(x)\|^2 + \frac{L}{2} \|M \nabla f(x)\|^2 \\
&\quad - (1 - \tau L) \langle \nabla f(x), M \nabla f(x) \rangle \\
&\leq f(x) - \tau \left(1 - \frac{L\tau}{2}\right) \|\nabla f(x)\|^2 + \frac{L\nu^2}{2} \|\nabla f(x)\|^2 + \nu |1 - \tau L| \|\nabla f(x)\|^2 \\
&= f(x) - \left( \tau \left(1 - \frac{\tau L}{2}\right) - \frac{L\nu^2}{2} - \nu |1 - \tau L| \right) \|\nabla f(x)\|^2 \\
&= f(x) - c(\nu, L, \tau) \|\nabla f(x)\|^2 ,
\end{aligned}
$$

where

$$
\begin{aligned}
c(\nu, L, \tau) &= \tau \left(1 - \frac{\tau L}{2}\right) - \frac{L\nu^2}{2} - \nu |1 - \tau L| \\
&= \frac{L}{2} \left( \frac{1}{L} - \nu - \left| \frac{1}{L} - \tau \right| \right) \left( \nu + \left| \frac{1}{L} - \tau \right| + \frac{1}{L} \right).
\end{aligned}
$$

Therefore

$$f(x_{t+1}) \leq f(x_t) - c(\nu, L, \tau) \|\nabla f(x)\|^2 . \tag{49}$$

Note that $\nu + \left| \frac{1}{L} - \tau \right| + \frac{1}{L} > 0$, so for $c(\nu, L, \tau)$ to be positive, we require

$$\frac{1}{L} - \nu - \left| \frac{1}{L} - \tau \right| > 0. \tag{50}$$

**Case 1** - $\frac{1}{L} \geq \tau$
Then we require

$$\tau - \nu > 0 \iff \nu < \tau, \tag{51}$$

which is true as we take $\nu \in [0, \tau)$.
**Case 2** - $\frac{1}{L} < \tau$
Then we require

$$\frac{2}{L} - \nu - \tau > 0 \iff \frac{1}{L} > \frac{\tau + \nu}{2}. \tag{52}$$

To conclude both cases, we have $\nu < \tau$ and therefore as $\frac{1}{\tau} < \frac{1}{\tau + \nu}$, we require only case 2 to be satisfied for $c(\nu, L, \tau) > 0$:

$$L < \frac{2}{\tau + \nu}. \tag{53}$$

In particular, any $L \leq L_{\text{train}}$ satisfies this inequality for any $\nu \in [0, \tau)$. $\qquad\square$

**Proposition 6.** *Assume that $G : \Theta \to \mathcal{L}(\mathcal{X})$ is continuous. Then at any iteration $t$ there exists $\lambda_t \geq 0$ and a constant $\tilde{L} > 0$ such that for all $f \in \mathcal{F}_{\tilde{L}}$ and any starting point $x_0$, using Algorithm 2 gives $\nabla f(x_t) \to 0$ as $t \to \infty$.*

*Proof.* Take $\nu \in (0, \tau)$. Define $h(\lambda) = \|G_{\arg\min_\theta g_{t,\lambda}(\theta)} - \tau I\| - \nu$ for $g_{t,\lambda}(\theta)$ as in (6). Note that $\lim_{\lambda \to \infty} h(\lambda) = -\nu < 0$. If $h(0) < 0$ then we are done as for $\lambda_t = 0$, the corresponding learned parameters $\theta_t$ satisfy $\|G_{\theta_t} - \tau I\| < \nu$, leading to a provably convergent algorithm for $f \in \mathcal{F}_{\tilde{L}}$ for some $\tilde{L} > 0$. Else, suppose that $h(0) > 0$. Then as $h$ is continuous in $\lambda$, there exists some $\lambda$ such that $h(\lambda) < 0$. $\square$

At the final training iteration $T$, to find a $\lambda_T$ that is large enough to ensure convergence, we start at an initial point $\lambda = 10^{-6}$ and find $\phi \in \arg\min_\theta g_{T,\lambda}(\theta)$. If $\|G_\phi - \tau I\| < \tau$, then increase $\lambda$ by a multiple and re-evalute. Repeat until this inequality no longer holds, and take $\lambda_T$ to be the most recent $\lambda$ such that $\|G_\phi - \tau I\| < \tau$. Else if $\lambda = 10^{-6}$ and $\phi \in \arg\min_\theta g_{T,\lambda}(\theta)$ satisfies $\|G_\phi - \tau I\| > \tau$ then reduce $\lambda$ by a multiple and re-evaluate until $\|G_\phi - \tau I\| < \tau$, then take $\lambda_T = \lambda$. For the (PS) parametrization we take the multiple to be 5, and for the (PP) parametrization, we take this multiple to be 2.

***Proof of Theorem 3.*** Define $D = \max_{t=0,1,\ldots}\{\|x_t - x^*\|\}$, which is finite as $(x_t)$ is bounded. Due to the convexity of $f$ and the Cauchy-Schwarz inequality, we have that

$$\begin{aligned} f(x_t) - f(x^*) &\leq \langle \nabla f(x_t), x_t - x^* \rangle \\ &\leq \|\nabla f(x_t)\| \|x_t - x^*\| \\ &\leq D\|\nabla f(x_t)\|. \end{aligned}$$

Therefore

$$\|\nabla f(x_t)\|^2 \geq \frac{1}{D^2}(f(x_t) - f(x^*))^2, \tag{54}$$

and for $t \geq T$ we have

$$\begin{aligned} f(x_{t+1}) &\leq f(x_t) - c(\nu, L, \tau)\|\nabla f(x_t)\|^2 \\ &\leq f(x_t) - \frac{c(\nu, L, \tau)}{D^2}(f(x_t) - f(x^*))^2. \end{aligned}$$

Denote $\Delta_t = f(x_{t+T}) - f(x^*)$, then in the spirit of (Nesterov et al., 2018), we have for all $t \geq 0$

$$\Delta_{t+1} \leq \Delta_t - \frac{c}{D^2}\Delta_t^2$$

$$\implies \frac{1}{\Delta_t} \leq \frac{1}{\Delta_{t+1}} - \frac{c}{D^2}\frac{\Delta_t}{\Delta_{t+1}} \leq \frac{1}{\Delta_{t+1}} - \frac{c}{D^2}$$

$$\implies \frac{c}{D^2} + \frac{1}{\Delta_t} \leq \frac{1}{\Delta_{t+1}}.$$

Taking a summation gives

$$\sum_{k=0}^{t-1} \frac{c}{D^2} \leq \sum_{k=0}^{t-1} \left( \frac{1}{\Delta_{k+1}} - \frac{1}{\Delta_k} \right)$$

$$\implies \frac{c}{D^2}t \leq \frac{1}{\Delta_t} - \frac{1}{\Delta_0}.$$

Therefore

$$\Delta_t \leq \frac{1}{\frac{1}{\Delta_0} + \frac{c}{D^2}t} = \frac{D^2\Delta_0}{D^2 + c\Delta_0 t} \leq \frac{D^2\Delta_0}{c\Delta_0 t} = \frac{D^2/c}{t},$$

as required.

$\square$

# E  PROOFS FOR SECTION 5

***Proof of Lemma 2.***     1. For scalar step sizes, $G_\theta = \theta I$, take $\tilde\theta = \tau$.

2. For a pointwise parametrization, $G_\theta x = \theta \odot x$, take $\tilde\theta = \tau \mathbf{1}$.

3. For full operator parametrization, $G_\theta = \theta \in \mathcal{L}(\mathcal{X})$, take $\tilde\theta = \tau I$.

4. For the convolutional parametrization, $G_\theta x = \theta * x$, take

$$\theta(i,j) = \begin{cases} \tau, & \text{if } i = j = 0, \\ 0, & \text{otherwise.} \end{cases} \tag{55}$$

$G_\theta$ are clearly continuous in $\theta$ for all listed parametrizations. $\qquad\square$

***Proof of Corollary 1.*** With any parametrization in Table 1, $G : \Theta \to \mathcal{L}(\mathcal{X})$ is continuous by Lemma 2. For Theorem 2 to hold, we then need $\theta_t$ is BGD and $\liminf_{t\to\infty} \lambda_t > 0$, which are both assumed. For Theorem 3, we only further require $(x_t)_{t=1}^\infty$, which is also assumed. $\qquad\square$

***Proof of Proposition 3.*** Because this problem is convex, if a solution $\theta$ is found by differentiating the objective function and equating equal to zero, this is a global minimizer. First, note that

$$\begin{aligned} f_k(x_k^t - B_k^t\theta) &= \frac{1}{2}\|A_k(x_k^t - B_k^t\theta) - y_k\|^2 \\ &= \frac{1}{2}\|A_k x_k^t - y_k\|^2 + \frac{1}{2}\| - A_k B_k^t\theta\|^2 + \langle -A_k B_k^t\theta, A_k x_k^t - y_k\rangle \\ &= \frac{1}{2}\|A_k x_k^t - y_k\|^2 + \frac{1}{2}\|A_k B_k^t\theta\|^2 - \langle\theta, (B_k^t)^*\nabla f_k(x_k^t)\rangle. \end{aligned}$$

Now,

$$\nabla_\theta \left\{ \frac{1}{N}\sum_{k=1}^N f_k(x_k^t - B_k^t\theta) + \frac{\lambda_t}{2}\|\theta - \tilde\theta\|^2 \right\}$$

$$= \frac{1}{N}\sum_{k=1}^N (A_k B_k^t)^*(A_k B_k^t\theta) - (B_k^t)^*\nabla f_k(x_k^t) + \lambda_t(\theta - \tilde\theta)$$

is equal to zero if and only if

$$\left(\frac{1}{N}\sum_{k=1}^N (A_k B_k^t)^*(A_k B_k^t) + \lambda_t I_\Theta\right)\theta = \lambda_t\tilde\theta + \frac{1}{N}\sum_{k=1}^N (B_k^t)^*\nabla f_k(x_k^t).$$

$\qquad\square$

**A bonus proposition regarding the uniqueness of optimal parameters.**

**Proposition 7.** $g_{t,\lambda_t}(\theta)$ *has a unique global minimizer $\theta_t^*$ if at least one of the following are satisfied:*

- $\lambda_t > 0$,

- $f_k$ *is twice continuously differentiable for $k \in \{1, \cdots, N\}$, and there exists some $j \in \{1, \cdots, N\}$ for which both $B_j^t$ is injective and also $f_j$ is $\mu_j$-strongly convex.*

*Proof.* **Case 1 -** $\lambda_t > 0$
$\frac{1}{N}\sum_{k=1}^N (A_k B_k^t)^*(A_k B_k^t)$ is self-adjoint and positive semi-definite as it is the sum of self-adjoint operators, $\frac{1}{N}\sum_{k=1}^N (A_k B_k^t)^*(A_k B_k^t) + \lambda_t I$ is a self-adjoint, positive-definite operator and therefore invertible.
**Case 2 -** $\lambda_t = 0$

If each $f_k$ is twice continuously differentiable; then $g_{t,\lambda_t}$ is twice continuously differentiable. It is then sufficient to show there exists $m > 0$ such that

$$\nabla^2 g_{t,\lambda_t}(\theta) \succeq mI, \tag{56}$$

for all $\theta$, as this implies that $g_{t,\lambda_t}$ is strongly convex and has a unique global minimizer. Note that

$$\nabla^2 g_{t,\lambda_t}(\theta) = \frac{1}{N}\sum_{k=1}^{N}(B_k^t)^*\nabla^2 f_k(x_k^t - B_k^t\theta)B_k^t. \tag{57}$$

Note that

$$\langle v, \nabla^2 g_{t,\lambda_t}(\theta)v\rangle = \langle v, \frac{1}{N}\sum_{k=1}^{N}(B_k^t)^*\nabla^2 f_k(x_k^t - B_k^t\theta)B_k^t v\rangle \tag{58}$$

$$= \frac{1}{N}\sum_{k=1}^{N}\langle v, (B_k^t)^*\nabla^2 f_k(x_k^t - B_k^t\theta)B_k^t v\rangle \tag{59}$$

$$= \frac{1}{N}\sum_{k=1}^{N}\langle B_k^t v, \nabla^2 f_k(x_k^t - B_k^t\theta)B_k^t v\rangle. \tag{60}$$

$$\tag{61}$$

Each $f_k$ is convex and so for all $v \in \mathcal{X}$,

$$\langle v, \nabla^2 f_k(x_k^t - B_k^t\theta)v\rangle \geq 0, \tag{62}$$

and $f_j$ is $\mu_j$-strongly convex, therefore

$$\langle v, \nabla^2 f_j(v_t^j - B_j^t\theta)v\rangle \geq \mu_j\|v\|^2. \tag{63}$$

For $v \in \mathcal{X}$,

$$\langle v, \nabla^2 g_{t,\lambda_t}(\theta)v\rangle \geq \frac{1}{N}\mu_j v^T(B_j^t)^*B_j^t v$$

$$\geq \left(\frac{1}{N}\mu_j\rho_{\min}^j\right)\|v\|^2,$$

where $\rho_{\min}^j$ is the minimum eigenvalue of $M_j^t = (B_j^t)^*B_j^t$ (a symmetric linear operator). Due to the symmetry of $M_j^t$, $\rho_{\min}^j \geq 0$ and is greater than zero if and only if $B_j^t$ is injective. As $B_j^t$ is injective, then $\rho_{\min}^j > 0$ and therefore $g_{t,\lambda_t}(\theta)$ is strongly-convex. $\square$

**Proposition 7 applied to least-square functions.**

**Corollary 2.** *Uniqueness of optimal parameters in the least-squares case*
*When our $f_k$ can be written as least-squares functions $f_k(x) = \frac{1}{2}\|A_k x - y_k\|^2$, then $g_{t,\lambda_t}(\theta)$ has a unique global minimizer $\theta_t^*$ if at least one of the following are satisfied:*

- *$\lambda_t > 0$,*

- *there exists some $j \in \{1, \cdots, N\}$ for which both $B_j^t$ and $A_j$ are injective.*

*Proof.* If $A_j$ is injective then $A_j^*A_j$ is invertible which means that $f_j(x) = \frac{1}{2}\|A_jx - y^j\|^2$ is strongly convex. $\square$

**Proposition 8.** $p_t$ *given by*

$$p_t = \left(\lambda_t I_\Theta + \frac{1}{N}\sum_{k=1}^{N}\left(\nabla f_k(x_k^t) \otimes \nabla f_k(x_k^t)\right) \odot (A_k^*A_k)\right)^{\dagger}\left(\lambda_t\tilde{\theta} + \frac{1}{N}\sum_{k=1}^{N}\nabla f_k(x_k^t) \odot \nabla f_k(x_k^t)\right) \tag{64}$$

*is a solution to (6) with the pointwise parametrization $G_{p_t}x = p_t \odot x$ for any $x \in \mathcal{X}$.*

*Proof.* Define for $x \in \mathcal{X}$,

$$B_k^t x = \nabla f_k(x_k^t) \odot x$$

then, for $x \in \mathcal{X}$, $(B_k^t)^*(x) = B_k^t(x)$. Now,

$$(A_k B_k^t)^*(A_k B_k^t)p = (B_k^t)^* \left( A_k^* A_k B_k^t p \right)$$
$$= \nabla f_k(x_k^t) \odot \left( A_k^* A_k(\nabla f_k(x_k^t) \odot p) \right).$$

Now,

$$[\nabla f_k(x_k^t) \odot \left( A_k^* A_k(\nabla f_k(x_k^t) \odot p) \right)]_j$$
$$= [\nabla f_k(x_k^t)]_j [A_k^* A_k(\nabla f_k(x_k^t) \odot p)]_j, \text{ by (15)}$$
$$= [\nabla f_k(x_k^t)]_j \sum_{i=1}^n [\nabla f_k(x_k^t) \odot p]_i [A_k^* A_k]_{ji}, \text{ by (20)}$$
$$= \sum_{i=1}^n [\nabla f_k(x_k^t)]_i [p]_i [\nabla f_k(x_k^t)]_j [A_k^* A_k]_{ji}, \text{ by (15).}$$

Secondly,

$$\left[ \left( \left( \nabla f_k(x_k^t) \otimes \nabla f_k(x_k^t) \right) \odot (A_k^* A_k) \right) p \right]_j$$
$$= \sum_{i=1}^n [p]_i [\left( \nabla f_k(x_k^t) \otimes \nabla f_k(x_k^t) \right) \odot (A_k^* A_k)]_{ji}, \text{ by (20)}$$
$$= \sum_{i=1}^n [p]_i [\nabla f_k(x_k^t) \otimes \nabla f_k(x_k^t)]_{ji} [A_k^* A_k]_{ji}, \text{ by (16)}$$
$$= \sum_{i=1}^n [p]_i [\nabla f_k(x_k^t)]_j [\nabla f_k(x_k^t)]_i [A_k^* A_k]_{ji} \text{ by (19)}$$
$$= [\nabla f_k(x_k^t) \odot \left( A_k^* A_k(\nabla f_k(x_k^t) \odot p) \right)]_j, \text{ by (18).}$$

Finally,

$$\lambda_t \tilde{\theta} + \frac{1}{N} \sum_{k=1}^N (B_k^t)^* \nabla f_k(x_k^t) = \lambda_t \tilde{\theta} + \frac{1}{N} \sum_{k=1}^N \nabla f_k(x_k^t) \odot \nabla f_k(x_k^t).$$

Then the result follows from proposition 3. $\square$

**Proposition 9.** *Let $B_k^t : \mathcal{L}(\mathcal{X}) \to \mathcal{X}$ be such that for any linear operator $P \in \mathcal{L}(\mathcal{X})$, we have $B_k^t(P) = P \nabla f_k(x_k^t)$. Then its adjoint $(B_k^t)^* : \mathcal{X} \to \mathcal{L}(\mathcal{X})$ is given by*

$$(B_k^t)^*(w) = \nabla f_k(x_k^t) \otimes w, \tag{65}$$

*for any element $w \in \mathcal{X}$. Then $\theta_t$ equal to*

$$\left( \lambda_t I_\Theta + \frac{1}{N} \sum_{k=1}^N (A_k^* A_k) \otimes (\nabla f_k(x_k^t) \otimes \nabla f_k(x_k^t)) \right)^\dagger \left( \lambda_t \tilde{\theta} + \frac{1}{N} \sum_{k=1}^N \nabla f_k(x_k^t) \otimes \nabla f_k(x_k^t) \right),$$
$$\tag{66}$$

*is a solution to (6) for the full operator parametrization.*

*Proof.* $\theta_t \in \mathcal{L}(\mathcal{X})$ and we require $\theta_t \nabla f_k(x_k^t) = B_k^t(\theta_t)$, so can take $B_k^t(\theta_t) = \theta_t \nabla f_k(x_k^t)$. For the adjoint,

$$\langle B_k^t(P), w \rangle = \sum_{i=1}^{n} [P \nabla f_k(x_k^t)]_i [w]_i \tag{67}$$

$$= \sum_{i=1}^{n} \sum_{j=1}^{n} [P]_{ij} [\nabla f_k(x_k^t)]_j [w]_i \tag{68}$$

$$= \langle P, (B_k^t)^* w \rangle \tag{69}$$

$$= \sum_{i=1}^{n} \sum_{j=1}^{n} [P]_{ij} [(B_k^t)^* w]_{ij}, \tag{70}$$

and therefore $[(B_k^t)^* w]_{ij} = w_i [\nabla f_k(x_k^t)]_j$, which means $(B_k^t)^*(w) = w \otimes \nabla f_k(x_k^t)$. Now,

$$[(A_k B_k^t)^*(A_k B_k^t)\theta]_{ij} = [(B_k^t)^*(A_k^* A_k B_k^t \theta)]_{ij}$$

$$= [(A_k^* A_k B_k^t \theta) \otimes \nabla f_k(x_k^t)]_{ij}$$

$$= [\nabla f_k(x_k^t)]_j [A_k^* A_k B_k^t \theta]_i, \text{ by (18)}$$

$$= [\nabla f_k(x_k^t)]_j \sum_{q=1}^{n} [A_k^* A_k]_{iq} [B_k^t \theta]_q$$

$$= [\nabla f_k(x_k^t)]_j \sum_{q=1}^{n} [A_k^* A_k]_{iq} \sum_{\ell=1}^{n} [\theta]_{q\ell} [\nabla f_k(x_k^t)]_\ell, \text{ (definition of } B_k^t).$$

Similarly,

$$[((A_k^* A_k) \otimes (\nabla f_k(x_k^t) \otimes \nabla f_k(x_k^t)))\theta]_{ij}$$

$$= \sum_{q,\ell=1}^{n} [(A_k^* A_k) \otimes (\nabla f_k(x_k^t) \otimes \nabla f_k(x_k^t))]_{ij,q\ell} [\theta]_{q\ell}$$

$$= \sum_{q,\ell=1}^{n} [A_k^* A_k]_{iq} [\nabla f_k(x_k^t) \otimes \nabla f_k(x_k^t)]_{j\ell} [\theta]_{q\ell}, \text{ by (19)}$$

$$= \sum_{q,\ell=1}^{n} [A_k^* A_k]_{iq} [\nabla f_k(x_k^t)]_j [\nabla f_k(x_k^t)]_\ell [\theta]_{q\ell}$$

$$= [(A_k B_k^t)^*(A_k B_k^t)\theta]_{ij},$$

as required, due to $[A_k^* A_k]_{jq} = [A_k^* A_k]_{qj}$.

$\square$

**Proposition 10.** *If each $f_k$ can be written as a least-squares function $f_k(x) = \frac{1}{2}\|A_k x - y_k\|^2$, then $\alpha_t$ can be given as*

$$\alpha_t = \frac{\lambda_t \tilde{\theta} + \frac{1}{N} \sum_{k=1}^{N} \|\nabla f_k(x_k^t)\|^2}{\lambda_t + \frac{1}{N} \sum_{k=1}^{N} \|A_k \nabla f_k(x_k^t)\|^2}, \tag{71}$$

*if $\lambda_t > 0$ or $A_j \nabla f_j(x_j^t) \neq \underline{0}$ for some $j \in \{1, \cdots, N\}$.*

*Proof.* Take $B_k^t : \mathbb{R} \to \mathcal{X}$ such that

$$B_k^t(\alpha) = \alpha \nabla f_k(x_k^t).$$

Then for $\alpha \in \mathbb{R}$

$$\langle B_k^t(\alpha), w \rangle = \langle \alpha \nabla f_k(x_k^t), w \rangle = \alpha \langle \nabla f_k(x_k^t), w \rangle.$$

Therefore

$$(B_k^t)^*(w) = \langle \nabla f_k(x_k^t), w \rangle. \tag{72}$$

then general formula 3 gives the desired result as

$$(B_k^t)^*(A_k^* A_k B_k^t(\alpha)) = \alpha \langle \nabla f_k(x_k^t), A_k^* A_k \nabla f_k(x_k^t) \rangle = \alpha \| A_k \nabla f_k(x_k^t) \|^2,$$
$$(B_k^t)^*(\nabla f_k(x_k^t)) = \langle \nabla f_k(x_k^t), \nabla f_k(x_k^t) \rangle = \| \nabla f_k(x_k^t) \|^2.$$

Then the result follows from proposition 3. $\square$

**Proposition 11.** *For $n_1, n_2 \in \mathbb{N}$, let $\mathcal{X} = \mathbb{R}^{n_1 \times n_2}$. Define $B_k^t : \mathcal{X} \to \mathcal{X}$ be such that for any convolutional kernel $\kappa \in \mathcal{X}$, we have $B_k^t(\kappa) = \kappa * \nabla f_k(x_k^t)$. Then its adjoint $(B_k^t)^* : \mathcal{X} \to \mathcal{X}$ is given by*

$$(B_k^t)^*(w) = w * \overline{\nabla f_k(x_k^t)}, \tag{73}$$

*where for $x \in \mathcal{X}$,*

$$\overline{x}(k, l) = x(-k, -l). \tag{74}$$

*Proof.* For the adjoint of $B_k^t$, we have

$$\langle B_k^t(\kappa), w \rangle = \langle \kappa * \nabla f_k(x_k^t), w \rangle \tag{75}$$

$$= \sum_{i,j} [\kappa * \nabla f_k(x_k^t)](i, j) w(i, j) \tag{76}$$

$$= \sum_{i,j} \sum_{k,l} \kappa(k, l) [\nabla f_k(x_k^t)](i - k, j - l) w(i, j) \tag{77}$$

$$= \sum_{i,j} \sum_{k,l} \kappa(k, l) [\nabla f_k(x_k^t)](i, j) w(i + k, j + l) \tag{78}$$

$$= \sum_{i,j} \sum_{k,l} \kappa(k, l) [\nabla f_k(x_k^t)](i, j) w(i + k, j + l) \tag{79}$$

$$= \sum_{i,j} [\nabla f_k(x_k^t)](i, j) \left( \sum_{k,l} \kappa(k, l) w(i + k, j + l) \right) \tag{80}$$

$$= \sum_{i,j} [\nabla f_k(x_k^t)](i, j) \left( \sum_{k,l} \kappa(-k, -l) w(i - k, j - l) \right) \tag{81}$$

$$= \langle \nabla f_k(x_k^t), \overline{\kappa} * w \rangle, \tag{82}$$

where $\overline{\kappa}(k, l) = \kappa(-k, -l)$. $\square$

### E.1 APPROXIMATING OPTIMAL LINEAR PARAMETERS

For general functions $f_k$, a closed-form solution does not exist for calculating linear parameters. Instead, we require an optimization algorithm to approximate these quantities. With information of $\nabla g_{t, \lambda_t}(\theta)$, and $L_{g_{t, \lambda_t}}$, the Lipschitz constant of $\nabla g_{t, \lambda_t}(\theta)$, one can use any first-order convex optimization algorithm, such as gradient descent, Nesterov accelerated gradient (Nesterov et al., 2018), or stochastic optimization methods such as SGD, and SVRG (Gower et al., 2020) (especially for large $N$, due to both speed and memory considerations) to approximate $\theta_t^*$. For example, one can start at an initial point $\theta_t^0$ at iteration $t$ and update via gradient descent

$$\theta_t^{w+1} = \theta_t^w - \frac{1}{L_{g_{t, \lambda_t}}} \nabla g_{t, \lambda_t}(\theta_t^w). \tag{83}$$

The following result illustrates how $\nabla g_{t, \lambda_t}(\theta)$ and $L_{g_{t, \lambda_t}}$ can be calculated.

**Proposition 12.** *For a general linear parametrization $G$, the gradient of $g_{t,\lambda_t}$ with respect to $\theta$ and its associated Lipschitz constant can be calculated as*

$$\nabla g_{t,\lambda_t}(\theta) = \lambda_t(\theta - \tilde{\theta}) - \frac{1}{N}\sum_{k=1}^{N}(B_k^t)^*\nabla f_k(x_k^t - G_\theta \nabla f_k(x_k^t)), \tag{84}$$

$$L_{g_{t,\lambda_t}} = \lambda_t + \frac{1}{N}\sum_{k=1}^{N}L_k\|B_k^t\|^2. \tag{85}$$

*Proof.* As

$$g_{t,\lambda_t}(\theta) = \frac{1}{N}\sum_{k=1}^{N}f_k(x_k^t - G_\theta \nabla f_k(x_k^t)) + \frac{\lambda_t}{2}\|\theta - \tilde{\theta}\|^2$$

$$= \frac{1}{N}\sum_{k=1}^{N}f_k(x_k^t - B_k^t\theta) + \frac{\lambda_t}{2}\|\theta - \tilde{\theta}\|^2,$$

then by the chain rule

$$\nabla g_{t,\lambda_t}(\theta) = -\frac{1}{N}\sum_{k=1}^{N}(B_k^t)^*\nabla f_k(x_k^t - B_k^t\theta) + \lambda_t(\theta - \tilde{\theta}), \tag{86}$$

as required. To calculate the smoothness constant, we have

$$\|\nabla g_{t,\lambda_t}(\theta_1) - \nabla g_{t,\lambda_t}(\theta_2)\|$$

$$= \left\|\lambda_t(\theta_1 - \theta_2) + \frac{1}{N}\sum_{k=1}^{N}(B_k^t)^*(\nabla f_k(x_k^t - B_k^t\theta_2) - \nabla f_k(x_k^t - B_k^t\theta_2))\right\|$$

$$\leq \lambda_t\|\theta_1 - \theta_2\| + \frac{1}{N}\sum_{k=1}^{N}\left\|(B_k^t)^*(\nabla f_k(x_k^t - B_k^t\theta_2) - \nabla f_k(x_k^t - B_k^t\theta_1))\right\|$$

$$\leq \lambda_t\|\theta_1 - \theta_2\| + \frac{1}{N}\sum_{k=1}^{N}\|B_k^t\|\left\|\nabla f_k(x_k^t - B_k^t\theta_2) - \nabla f_k(x_k^t - B_k^t\theta_1)\right\|$$

$$\leq \lambda_t\|\theta_1 - \theta_2\| + \frac{1}{N}\sum_{k=1}^{N}L_k\|B_k^t\|\left\|B_k^t(\theta_1 - \theta_2)\right\|$$

$$\leq \left(\lambda_t + \frac{1}{N}\sum_{k=1}^{N}L_k\|B_k^t\|^2\right)\|\theta_1 - \theta_2\|$$

Due to the properties of the triangle inequality, the Cauchy-Schwarz inequality, and the operator norm, this bound is tight. Therefore the Lipschitz constant of $\nabla g_{t,\lambda_t}(\theta)$ is given by

$$\lambda_t + \frac{1}{N}\sum_{k=1}^{N}L_k\|B_k^t\|^2 \tag{87}$$

as required. $\square$

Using this general result, we can calculate these values for specific parametrizations of $G$.

**Corollary 3.** *Suppose each $f_k \in \mathcal{F}_{L_k}$.*
***Pointwise parametrization***
*For the pointwise parametrization, $\theta \in \mathcal{X}$, and*

$$\nabla g_{t,\lambda_t}(\theta) = \lambda_t(\theta - \tilde{\theta}) - \frac{1}{N}\sum_{k=1}^{N}\nabla f_k(x_k^t - \theta \odot \nabla f_k(x_k^t)) \odot \nabla f_k(x_k^t), \tag{88}$$

*and an upper bound of the Lipschitz constant of $\nabla_\theta g$ is given by*

$$L_{\nabla_\theta g} = \lambda_t + \frac{1}{N} \sum_{k=1}^{N} L_k (\max\{|[\nabla f_k(x_k^t)]_1|, \cdots, |[\nabla f_k(x_k^t)]_n|\})^2. \tag{89}$$

**Full operator parametrization**
*In this case we have $\theta \in \mathcal{L}(\mathcal{X})$. The gradient of $g_{t,\lambda_t}(\theta)$ is given by*

$$\nabla g_{t,\lambda_t}(\theta) = \lambda_t(\theta - \tilde{\theta}) - \frac{1}{N} \sum_{k=1}^{N} \nabla f_k(x_k^t - \theta \nabla f_k(x_k^t)) \otimes \nabla f_k(x_k^t), \tag{90}$$

*and an upper bound of the Lipschitz constant of $\nabla g_{t,\lambda_t}(\theta)$ is given by*

$$\lambda_t + \frac{1}{N} \sum_{k=1}^{N} L_k \left\| \nabla f_k(x_k^t) \right\|^2. \tag{91}$$

**Scalar step size**
*We now take $\theta \in \mathbb{R}$. The derivative of $g_{t,\lambda_t}$ with respect to $\theta$ is given by*

$$g'_{t,\lambda_t}(\theta) = \lambda_t(\theta - \tilde{\theta}) - \frac{1}{N} \sum_{k=1}^{N} \langle \nabla f_k(x_k^t - \theta \nabla f_k(x_k^t)), \nabla f_k(x_k^t) \rangle, \tag{92}$$

*and the Lipschitz constant of $g'(\theta)$ is given by*

$$\lambda_t + \frac{1}{N} \sum_{k=1}^{N} L_k \|\nabla f_k(x_k^t)\|^2. \tag{93}$$

**Convolution**
*In this case we have $\theta \in \mathbb{R}^{n_1 \times n_2}$. The gradient of $g_{t,\lambda_t}(\theta)$ is given by*

$$\nabla g_{t,\lambda_t}(\theta) = \lambda_t(\theta - \tilde{\theta}) - \frac{1}{N} \sum_{k=1}^{N} \nabla f_k(x_k^t - G_\theta \nabla f_k(x_k^t)) * \overline{\nabla f_k(x_k^t)}. \tag{94}$$

*Proof.* **Pointwise parametrization**

In this case, we have $\theta \in \mathcal{X}$ and $B_k^t(x) = \nabla f_k(x_k^t) \odot x$ and $(B_k^t)^*(x) = B_k^t x$ for $x \in \mathcal{X}$. Furthermore,

$$\|B_k^t\| = \max_{x \neq 0} \frac{\|x \odot \nabla f_k(x_k^t))\|}{\|x\|} = \max_{x \neq 0} \sqrt{\frac{\sum_{i=1}^{n} [x]_i^2 [\nabla f_k(x_k^t)]_i^2}{\sum_{i=1}^{n} [x]_i^2}}$$

$$\leq \max_q |[\nabla f_k(x_k^t)]_q| \max_{x \neq 0} \sqrt{\frac{\sum_{i=1}^{n} [x]_i^2}{\sum_{i=1}^{n} [x]_i^2}} = \max\{|[\nabla f_k(x_k^t)]_1|, \cdots, |[\nabla f_k(x_k^t)]_n|\}.$$

**Full operator parametrization**

In the case of the full operator parametrization, we have $(B_k^t)^*(w) = w \otimes \nabla f_k(x_k^t)$. Therefore, using Proposition 12 gives (90). For the Lipschitz constant, note that

$$\|B_k^t(P)\| = \|P \nabla f_k(x_k^t)\| \leq \|P\| \|\nabla f_k(x_k^t)\|,$$

and therefore

$$\|B_k^t\| = \max_{P \neq 0} \frac{\|B_k^t(P)\|}{\|P\|} \leq \|\nabla f_k(x_k^t)\|.$$

**Scalar step size**

Let $B_k^t$ be defined for any $\alpha \in \mathbb{R}$ by $B_k^t(\alpha) = \alpha \nabla f_k(x_k^t)$, then for an element $w \in \mathcal{X}$, $(B_k^t)^*(w) = \langle \nabla f_k(x_k^t), w \rangle$. Furthermore,

$$\|B_k^t(\alpha)\| = \|\alpha \nabla f_k(x_k^t)\|$$
$$= |\alpha| \|\nabla f_k(x_k^t)\|,$$

and so

$$\|B_k^t\| = \max_{\alpha \neq 0} \frac{\|B_k^t(\alpha)\|}{|\alpha|} = \|\nabla f_k(x_k^t)\|.$$

**Convolution**

For the gradient,

$$(B_k^t)^*(\nabla f_k(x_k^t - G_\theta \nabla f_k(x_k^t))) = \nabla f_k(x_k^t - G_\theta \nabla f_k(x_k^t)) * \overline{\nabla f_k(x_k^t)}.$$

$\square$

For any chosen linear parametrization, one can approximate the operator norm of $B_k^t$ using the power method (Golub & Van Loan, 2013). The following table summarises the previous propositions:

Table 2: Example parametrization properties

| Parametrization | Equations |
|---|---|
| **Pointwise** | • $\tilde{\theta} = \tau \mathbf{1} \in \mathcal{X}$
• $B_k^t(x) = \nabla f_k(x_k^t) \odot x$, $(B_k^t)^*(x) = B_k^t(x)$
• $\nabla g_{t,\lambda_t}(\theta) = \lambda_t(\theta - \tilde{\theta}) - \frac{1}{N} \sum_{k=1}^N \nabla f_k(x_k^t - \theta \odot \nabla f_k(x_k^t)) \odot \nabla f_k(x_k^t)$
• $L_{g_{t,\lambda_t}} \leq \lambda_t + \frac{1}{N} \sum_{k=1}^N L_k(\max\{|\nabla f_k(x_k)_1|, \ldots, |\nabla f_k(x_k)_n|\})^2$ |
| **Full operator** | • $\tilde{\theta} = \tau I \in \mathcal{L}(\mathcal{X})$
• $B_k^t(P) = P \nabla f_k(x_k^t)$, $(B_k^t)^*(w) = w \otimes \nabla f_k(x_k^t)$
• $\nabla g_{t,\lambda_t}(\theta) = \lambda_t(\theta - \tilde{\theta}) - \frac{1}{N} \sum_{k=1}^N \nabla f_k(x_k^t - \theta \nabla f_k(x_k^t)) \otimes \nabla f_k(x_k^t)$
• $L_{g_{t,\lambda_t}} \leq \lambda_t + \frac{1}{N} \sum_{k=1}^N L_k \|\nabla f_k(x_k^t)\|^2$ |
| **Scalar** | • $\tilde{\theta} = \tau \in \mathbb{R}$
• $B_k^t(\alpha) = \alpha \nabla f_k(x_k^t)$, $(B_k^t)^*(w) = \langle w, \nabla f_k(x_k^t) \rangle$
• $g'_{t,\lambda_t}(\theta) = \lambda_t(\theta - \tilde{\theta}) - \frac{1}{N} \sum_{k=1}^N \langle \nabla f_k(x_k^t - \theta \nabla f_k(x_k^t)), \nabla f_k(x_k^t) \rangle$
• $L_{g_{t,\lambda_t}} = \lambda_t + \frac{1}{N} \sum_{k=1}^N L_k \|\nabla f_k(x_k^t)\|^2$ |
| **Convolution** | • $\tilde{\theta}(i,j) = \begin{cases} \tau, & \text{if } i = j = 0, \\ 0, & \text{otherwise.} \end{cases}$
• $B_k^t(\kappa) = \kappa * \nabla f_k(x_k^t)$, $(B_k^t)^*(\kappa) = \kappa * \overline{\nabla f_k(x_k^t)}$
• $g'_{t,\lambda_t}(\theta) = \lambda_t(\theta - \tilde{\theta}) - \frac{1}{N} \sum_{k=1}^N \nabla f_k(x_k^t - \theta * \nabla f_k(x_k^t)) * \overline{\nabla f_k(x_k^t)}$ |

# F    ADDITIONAL NUMERICAL RESULTS

## F.1    ABLATION STUDY: SIZE OF LEARNED KERNELS

Figure 8 shows that many of the learned convolutional algorithms outperform NAG for the deblurring problem. We see that the $5 \times 5$ kernels significantly outperform the NAG kernels and perform similarly to the $7 \times 7$ kernels. Furthermore, we see similar performance for the $11 \times 11$ kernels and the $96 \times 96$ kernels.

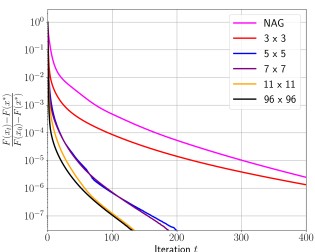

Figure 8: Test performance of different kernel sizes in the convolutional parametrisation, averaged over the test dataset in the deblurring problem. Tested kernel sizes are $3 \times 3$, $5 \times 5$, $7 \times 7$, $11 \times 11$, $3 \times 3$, $96 \times 96$.

### F.2 INVERSE KERNEL

For the operator $A$ given by a Gaussian blur with standard deviation $1.5$ and kernel size $5 \times 5$, and a constant $\delta = 0.2$, the operator $(\delta I + A^* A)^{-1}$ corresponds to a convolution with the kernel gives as in Figure 9.

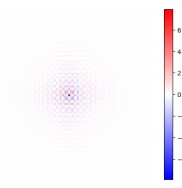

Figure 9: The kernel corresponding to the operator $(\delta I + A^* A)^{-1}$.

### F.3 SMALL-SCALE CT EXTRA RESULTS

**Visualising learned preconditioners.** Figure 10 shows that the learned scalar values again eventually fluctuate above and below $2/L$, similar to the behavior observed for the deblurring problem. The learned pointwise operators also exhibit oscillations between consecutive iterations, with many values falling outside the interval $(0, 2/L)$. Likewise, the learned convolutional kernels for the CT problem contain both positive and negative values, are predominantly weighted toward the center of the kernel, and become increasingly similar as the iterations progress.

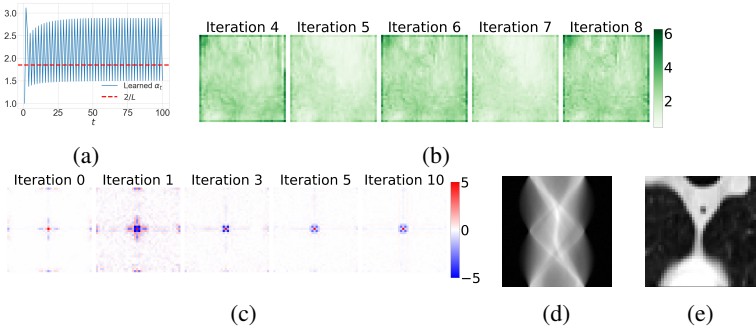

Figure 10: (a) Learned scalars for $t \in \{0, 1, \cdots, 100\}$. (b) Learned pointwise operators for $t \in \{4, 5, 6, 7, 8\}$. (c) Learned kernels restricted to the interval $[-5, 5]$, for $t \in \{0, 1, 3, 5, 10\}$. (d) Example CT observation. (e) Reconstruction by minimizing 13.

**Reconstruction comparison.** Figure 11 demonstrates that the learned parametrization (PC) achieves high-quality reconstructions with significantly fewer iterations compared to NAG for the small-

scale CT problem. Also, note that the regularized (PF) parametrization achieves a good visual reconstruction after only two iterations while also providing guaranteed convergence.

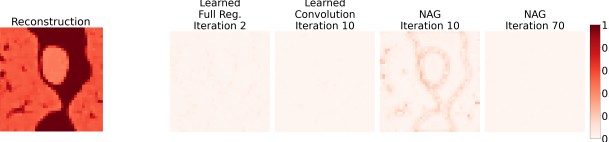

Figure 11: Left: An example reconstruction for the small-scale CT problem. Right: The absolute difference between the final reconstruction and the intermediate reconstruction for the full parametrization with regularization at iteration 2, the convolutional parametrization at iteration 10, and for NAG at iterations 10 and 70.

### F.4 LARGE-SCALE CT EXTRA RESULTS

In Figure 12 we see that the entire learned kernels for the large-scale CT problem are heavily weighted towards the center.

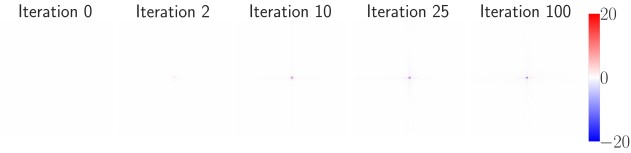

Figure 12: Learned kernels restricted to the interval $[-20, 20]$, for $t \in \{0, 2, 10, 25, 100\}$.

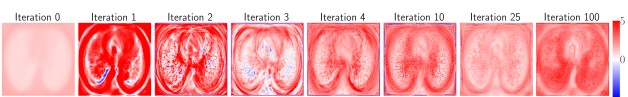

Figure 13: Learned pointwise operators restricted to the interval $[-5, 5]$, for $t \in \{0, 1, 2, 3, 4, 10, 25, 100\}$.

### F.5 TOLERANCE TABLES

Table 3: The first row shows error thresholds for the deblurring problem. The entries in the table show the number of required iterations to fall below the respective error threshold. "na" means that the threshold was not reached within 250 iterations for learned algorithms and 1000 iterations otherwise.

|  | $10^{-1}$ | $10^{-2}$ | $10^{-3}$ | $10^{-4}$ | $10^{-5}$ | $10^{-6}$ | $10^{-7}$ | $10^{-8}$ |
|---|---|---|---|---|---|---|---|---|
| Learned Convolution | 1 | 1 | 2 | 6 | 21 | 51 | 102 | 182 |
| NAG | 3 | 16 | 59 | 134 | 240 | 374 | 568 | 866 |
| L-BFGS | 3 | 15 | 51 | 130 | 253 | 416 | 599 | 892 |
| PGD | 2 | 5 | 58 | 263 | 878 | na | na | an |
| Learned Scalar | 3 | 30 | na | na | na | na | na | na |
| Backtracking GD | 3 | 31 | 308 | na | na | na | na | an |
| Learned Pointwise | 3 | 53 | na | na | na | na | na | na |

Table 4: The first row shows error thresholds for the large-scale CT problem. The entries in the table show the number of required iterations to fall below the respective error threshold. "na" means that the threshold was not reached within 200 iterations for learned algorithms and 1000 iterations otherwise.

| | $10^{-3}$ | $10^{-4}$ | $10^{-5}$ | $10^{-6}$ | $10^{-7}$ | $10^{-8}$ | $10^{-9}$ |
|---|---|---|---|---|---|---|---|
| Learned Convolution | 2 | 4 | 7 | 11 | 30 | 81 | 140 |
| NAG | 5 | 11 | 21 | 37 | 63 | 163 | 277 |
| L-BFGS | 4 | 9 | 19 | 36 | 64 | 142 | 304 |
| Backtracking GD | 6 | 16 | 49 | 135 | 354 | na | na |
| Learned Pointwise | 2 | 12 | 45 | 146 | na | na | na |
| Learned Scalar | 4 | 15 | 46 | 128 | na | na | na |

Table 5: The first row shows error thresholds for the small-scale CT problem. The entries in the table show the number of required iterations to fall below the respective error threshold. "na" means that the threshold was not reached within 200 iterations (or 100 in the case of the Full Regularized parametrization).

| | $10^{-3}$ | $10^{-4}$ | $10^{-5}$ | $10^{-6}$ | $10^{-7}$ | $10^{-8}$ | $10^{-9}$ |
|---|---|---|---|---|---|---|---|
| Learned Convolution | 2 | 4 | 6 | 9 | 12 | 17 | 25 |
| L-BFGS | 4 | 8 | 15 | 26 | 40 | 56 | 76 |
| NAG | 4 | 8 | 15 | 25 | 37 | 54 | 87 |
| Full Regularized | 1 | 2 | 2 | 17 | 88 | na | na |
| Backtracking GD | 6 | 13 | 28 | 68 | 136 | na | na |
| Learned Scalar | 3 | 9 | 24 | 59 | 115 | 190 | na |
| Learned Pointwise | 2 | 8 | 24 | 59 | 118 | 194 | na |

