# OpenReview forum: "Greedy Learning to Optimize with Convergence Guarantees"
_ICLR.cc/2025/Conference — Submitted to ICLR 2025_

### Official Review · Reviewer_6jvT · 2024-10-26

**Soundness:** 4
**Presentation:** 3
**Contribution:** 3
**Rating:** 8
**Confidence:** 4

**Summary:**

This work proposes a provably convergent learning-to-optimize method based on preconditioned gradient descent. By considering gradient descent and regularizing the proposed algorithm such that it is majorized by and eventually becomes GD, they demonstrate significant speedups on various convex optimization problems, while maintaining provable convergence guarantees from GD. Moreover, the linear parameterization allows for convex solvers at each timestep which gives speedup compared to other L2O methods.

**Strengths:**

* The paper reads well and the theorems are very intuitive. Overall, a good submission to ICLR.
* The experiments are convincing and the details are very comprehensive. The speed of training is especially good due to the simple parameterization, and comparisons are made with the greedy training vs standard unrolled training.
* The clear structure of the learned kernels (e.g. Figure 1c, 5c) is good cause for further investigation towards optimal preconditioning for certain imaging problems.

**Weaknesses:**

* The theory seems to require the knowledge of the maximum Lipschitz constant over all training examples in its regularization. How well does the method work when applied to problems with large or unknown constant?
* Related: finding the regularization parameter $lambda_T$ as in p.18 seems arduous. Is this done at every iteration $t$, and how many problems need to be solved to find $\lambda_t$? It seems also that the condition for choosing $\lambda_t$ gives that $G_\phi$ is positive definite, which is confusing with the initial claim that the method converges for non p.d. conditioners.
* While stepsize $\tau = 1/L$ does indeed give convergence for NAG, in practice, the smoothness is unknown and larger step-sizes can be taken while maintaining empirical convergence. Same for L-BFGS. Since the convergence profiles are quite similar, a proper comparison with differing parameters for NAG and L-BFGS would greatly help in ascertaining the role of interacting pixels in helping optimization.

**Questions:**

* (l.327) Should this be (PC) instead of (PP)?
* How are the ground truths $x^*$ computed?
* While the linear parameterization should give minimal overhead, a wall-clock time comparison of test-time might be useful for further comparison.
* For quadratic problems, Tan et al. (2023b) consider the preconditioning $G = (A^* A)^{-1}$, which can be used to motivate the convolutional structure. How does the proposed method compare to perhaps a regularized version, say $(I + \delta^{-1} A^* A)^{-1} \approxeq I - \delta A^\dagger A^{*\dagger}$. I am not sure if this is available in standard libraries.
* The convolution-based preconditioner seems to generalize something called "Laplacian smoothing gradient descent" (Osher et al., 2022). Have the authors considered other possible instances of such linear preconditionings that may have better empirical performance?
* Related: perhaps a short reference to App. D.2 would be helpful in the main text, as the choice of full-image convolution is not motivated.
* (l.737) RHS of first inequality should be $g_{t,0}(\tilde{\theta})$, and $\nabla f_k$ on the second line. Proof of Thm 1 perhaps needs a short clarification on telescoping arguments (relating objective of $x_{t+1}$ in terms of $x_t$) since $x_t$ is not generated using GD, but the result should be the same.
* Prop 4 can be more proved more succinctly by noting the residual of $F$ is the average of residuals of $f_k$, which are non-negative, so $f_k^t - f_k^* \le N F_t - F^*$.
* Prop 5 can directly use coordinate projection and remove the elementary calculation: consider the convex function $\pi_i f$ and first-order optimality, clearly minimized at $x^*$ and with derivative equal to $[\nabla f]_i$.
* Proof of Lem 1: notation from $g_{t,\lambda_t}$ to $g_t(\cdot, \lambda_t)$. Inconsistency in the first inequality with definition of BGD: should be $g_t(\theta_t, \lambda_t) \le g_t(\tilde{\theta}, 0)$.
* Line 40: perhaps reference Theorem 1 here when claiming that the sums of the $f_k$ converge to the optimal values.

[1] Osher, S., Wang, B., Yin, P., Luo, X., Barekat, F., Pham, M., & Lin, A. (2022). Laplacian smoothing gradient descent. Research in the Mathematical Sciences, 9(3), 55.

---

> ### Author Response · Authors · 2024-11-21
>
> We greatly appreciate the reviewer's detailed suggestions for improvement and positive feedback, thank you.
>
> >  ### “The theory seems to require the knowledge of the maximum Lipschitz constant over all training examples in its regularization”:
>
> We agree that the theory requiring knowledge of the maximum Lipschitz constant could be a challenge in practical applications where this constant is unknown. The method currently hasn’t been tested on problems with unknown Lipschitz smoothness.
>
> However, convergence on training data can be extended to Lipschitz smooth functions with an unknown constant (if the regularization function $\lambda_t R(\tilde{\theta})$ is removed), by instead proving convergence using a comparison to backtracking line search $F(x_t) - F(x_{t+1}) \geq \alpha h_t \|\nabla F(x_t) \|^2$ for some $\alpha \in (0,1)$ and a “small enough” step size $h_t$.
>
> Similarly, exact knowledge of the Lipschitz constant is not needed as the regularizer can be chosen to bias towards any parameters such that Gradient Descent is convergent.
>
> > ### “finding the regularization parameter $\lambda_T$ seems arduous”:
>
> We agree that finding the regularization parameter $\lambda_T$ can indeed be arduous if done so as described on page 18. In our experiments, either $\lambda_t$ is constant over time (In the case of the regularized full operator in Figure 6), or $\lambda_t = 0$ for all $t<T$, then we calculate $\lambda_T$ as in page 18. We have updated lines 929-934 to replace $\lambda_t$ with $\lambda_T$ to make this more explicit.
>
> > ### “ It seems also that the condition for choosing \lambda_t gives that $G_{\phi}$ is positive definite, which is confusing with the initial claim that the method converges for non p.d. conditioners.”
>
> To clarify, on the training dataset, we achieve convergence for $t \to \infty$ even when $\lambda_t = 0$ for all $t$. This means that the preconditioners need not be SPD. But it is true that when training is terminated at some iteration $T$ we need this final preconditioner to be positive definite (we have $G_{\theta_T} = \tau I + M$ with $\|M\| \leq \nu < \tau$, meaning that $x^TG_{\theta_T}x = \tau \|x\|^2 + x^TMx \geq \tau \|x\|^2 - \|M\|\|x\|^2 > 0$).
>
> > ### “How are the ground truths x^* computed?”
>
> The problems we solve are toy problems, and we have access to ground-truth images (e.g. clean images, denoted by $x_{\text{true}}$ in the paper), then generate observations y by applying the forward operator A and Gaussian noise. From then we only use information of the function $f(x) = \frac12 \|Ax-y\|^2 + \alpha H_{\epsilon}(x)$ and the initialisation $x_0$.
> In our paper, $F(x^*)$ is used to denote the minimum value of the function $F$ defined in line 330, which is approximated as we do not have access to the exact minimiser of F. This approximation is used as $F(x^*)$ in Figures 2, 4 and 6.
>
> We would like to thank the reviewer for noticing many other improvements to the submission.

---

> > ### Comment · Reviewer_6jvT · 2024-11-24
> >
> > I thank the authors for their feedback regarding my questions. I would suggest adding a short paragraph stating how the minimum value of $F$ is approximated, presumably by some gradient method. I have no further questions and maintain my score.

---

### Official Review · Reviewer_yL8M · 2024-11-01

**Soundness:** 3
**Presentation:** 3
**Contribution:** 3
**Rating:** 6
**Confidence:** 3

**Summary:**

This paper proposed a novel L2O approach for inverse problems. To mitigate the computational challenges in current approaches in L2O, which are mostly based on unrolling, this paper investigate a greedy training approach which decouples the iterates leading to more efficient training. The proposed scheme provides an effective approach for training a preconditioner for gradient descent. Theoretical analysis demonstrate that the proposed preconditioned gradient descent converges under the BGD condition. A specialized parameterization using convolution has been proposed in the paper which is tailored for imaging applications. Numerical experiments on inverse problems have demonstrated superior performance over classical methods such as Nesterov's accelerated gradient and L-BFGS.

**Strengths:**

The greedy training approach is a novel and interesting scheme for L2O. Indeed, current L2O methods are mostly depending on unrolling several iterations, the proposed scheme is very timely for L2O area for scalability of training the optimizer.

The numerical performance of the proposed scheme on inverse problems is very impressive.

**Weaknesses:**

The theoretical part of the paper seems to be weak. The convergence analysis is relying on an unrealistic assumption named BGD (better than gradient descent) assumption across each iteration -- you can't just simply assume what you wish to proof. Corollary 1 seems to give a very strong claim but no explicit proof is given (it is unclear how simply applying Lemma 2 can lead to such claim). The reviewer believes that whether or not the learned linear preconditioner is BGD should be non-trivial to show -- it should certainly depend on the actual values of the trained parameters. The "with Convergence Guarantees" part of the claimed contribution is unfortunately not valid.

In terms of further development, a limitation of the proposed scheme is inability to learn those iterative schemes which utilize memory of past iterates -- that is, in fact an advantage of unrolling which deserves acknowledgement. The reviewer believes that, ultimately, the proposed greedy scheme should be jointly applied with unrolling for the best performance.

In terms of experiments, although classical hand-crafted optimizers are included as baselines, there is no comparison with other existing L2O methods. For example, the author(s) could consider the SOTA (truly) provably convergent method by: Banert S, Rudzusika J, Öktem O, Adler J. Accelerated forward-backward optimization using deep learning. SIAM Journal on Optimization. 2024 Jun 30;34(2):1236-63.

**Questions:**

As mentioned above, please clarify the doubts regarding the theoretical analysis and include comparision with existing provable L2O schemes.

Meanwhile, the proposed scheme is tailored for inverse problems in imaging, particularly the convolutional parameterization of the preconditioner -- should this be make clearer in title/abstract/introduction?

---

> ### Author Response · Authors · 2024-11-21
>
> We would first like to thank the reviewer for their constructive feedback. Specific comments are addressed below.
>
> > ### “an unrealistic assumption named BGD”:
>
> The BGD assumption is saying that $g_{t, \lambda_t} (\theta_t) \leq g_{t, \lambda_t} (\tilde{\theta})$ where $\tilde{\theta}$ are the parameters that correspond to Gradient Descent.
>
> We learn $\theta_t = \arg\min_{\theta} g_{t, \lambda_t} (\theta)$, then this value would automatically satisfy the BGD property if our parametrizations generalise Gradient Descent.  It is easy to check that this is satisfied for all considered examples.
>
> Note also that this is not very restrictive and it is simple to modify any other given parametrization to have this property.
>
> In practice, of course, the minimization over the parameters is not solved exactly but just an approximation. However, the BGD property is verified in training by calculating and comparing $g_{t, \lambda_t} (\theta_t)$ and $g_{t, \lambda_t} (\tilde{\theta})$, and is always found to hold in our numerical experiments.
>
> > ### “Corollary 1 seems to give a very strong claim but no explicit proof is given”:
>
> Theorem 2 requires
> 1. $G_{\theta}$ is continuous with respect to $\theta$.
> 2. $\theta_t$ is BGD
> 3. $\lim \inf \lambda_t$ > 0.
>
> In the statement of corollary 1, we assume points 2 and 3 so all that is left to prove is point 1, which is proved in Lemma 2. The proof of corollary 1 has been extended to what is contained in this explanation.
>
> Similar to other proofs in optimization, we also require $(x_t)_{t=1}^\infty$ to be bounded for Theorem 3 to hold. This explanation has been added to the paper.
>
> > ### "a limitation of the proposed scheme is inability to learn those iterative schemes which utilize memory of past iterates"
>
> We agree with the reviewer that this is a limitation. However, our approach is able to learn over hundreds or thousands of iterations due to this and still one obtains excellent empirical performance, so it can be seen as a tradeoff.

---

> > ### Comment · Reviewer_yL8M · 2024-11-25
> >
> > Thanks for the clarification

---

### Official Review · Reviewer_aeaw · 2024-11-03

**Soundness:** 2
**Presentation:** 2
**Contribution:** 2
**Rating:** 5
**Confidence:** 3

**Summary:**

The submission introduces a "learning to optimize" algorithm that learns a sequence of fixed preconditioners to apply to gradient descent by fitting them greedily to maximize the one-step progress on a set of training problems. The paper presents convergence guarantees that the algorithm can fit and is guaranteed to converge when run for longer than trained on some problems that are not in the training set. The submission introduces multiple types of preconditioners and shows experimental results on deblurring and tomography problems.

**Strengths:**

The learning to optimize literature is still in development, and does not yet have well defined formalism or benchmark problems. As such, the goal of the paper as bringin formal guarantees to this setting is a new and relevant contribution to the community. The proposed algorithms show promising results on the experimental setup.

**Weaknesses:**

**Safeguards.** The proposed algorithm still relies on hand-crafted a-priori knowledge of the optimization problem in the form of the maximum step-size that would work for all training problem, $\tau$. As such, it does not significantly differ from alternative approaches that require safeguarding or search within a predefined set that guarantees convergence.

**Experimental validation.** The definition of train and test problems in the numerical experiments is not sufficiently transparent. If I understood it correctly, both problems are of the form $\|Ax-b\|^2$ but the training and test "samples" only differ in $x$ and $b$; the linear operator $A$ is fixed. This seems like an ``easy'' problem, as the goal of learning algorithm is to learn a preconditioner that approximates the inverse of $A^TA$, which is fixed across both training and test. The paper should make the distinction clear and highlight possible limitations. A more thorough experimental evaluation that tests the trained algorithm against other blurring operators, at least Gaussian blur with different parameters, would help make the claim that the learning algorithm can indeed generalize.

**Generalization guarantees.** The submission claims that the given algorithm "ensures convergence on unsees data" but the guarantees seem weak. Theorem 2 guarantees that there exists functions for which the given algorithm will work, but this is weaker than typical generalization guarantee. Translated to the classification setting, I would take to mean "there exists samples that were not in the training set that the algorithm classifies correctly", which is not a strong statement about the performance of the algorihm. Would it be possible to guarantee that the algorithm would work on all $1/\tau$ smooth problems, or to characterize that the preconditioner learned by the algorithm will lead to better performance on other problems if the train and test problems use the same operator $A$?

**Convolutional preconditioners** That the experimental results show a significant improvement in performance when used the learned convolutional preconditioner makes it unclear whether the benefit arises from the convolutional parameterization or the "learning to optimize" approach, and one could envision a BFGS-like algorithm using the convolutional structure. Although this is not my area of expertise, my understanding is that specialized approaches for deblurring using convolutional preconditioners exist, see for example [the work of Eboli et at.](https://arxiv.org/pdf/2007.01769). A discussion of, and a comparison with, specialized algorithms would be a welcome addition.

**Questions:**

## Smaller concerns

- **BGD assumption** Please correct me otherwise, but the "Better than Gradient Descent" (BGD) is assumed rather rather than proved, and theorem 2 both uses that $t \to \infty$ while requiring a final training iteration $T$. These two statements seem contradictory?
- The BGD assumption seem unecessary for the unseen problem proof? A similar argument could be made without the $t \to \infty$ or BGD assumption if the preconditioner is PD. For a given training budget, the algorithm repeats the last learned preconditioner $G_{\theta_{T-1}}$, so it will eventually converge on any unseen function that is smooth relative to $G_{\theta_{T-1}}$ in the sense that $\nabla^2 f(x) \preceq [G_{\theta_{T-1}}]^{-1}$. This doesn't seem much weaker than the current proposition, which only guarantees that there exists a smoothness constant $\tilde L$ such that the algorithm will convergence on $\tilde L$-smooth functions.

- **Related work in optimization**: The discussion of related work in optimization only touches on L-BFGS and ignores relevant work that attempt to achieve a similar goal, "find a better step-size/preconditioner for the problem", but by running additional computation before/while solving the problem rather than by taking the learning to optimize approach. While the approaches are different, this literature should at least be acknowledged in a paragraph in the introduction as it shares a similar goal. Examples include a simple [Armijo line-search](https://projecteuclid.org/journals/pacific-journal-of-mathematics/volume-16/issue-1/Minimization-of-functions-having-Lipschitz-continuous-first-partial-derivatives/pjm/1102995080.full),
[optimal diagonal preconditioners](https://arxiv.org/abs/2209.00809) for quadratic,
[AdaGrad](https://www.jmlr.org/papers/volume12/duchi11a/duchi11a.pdf) and [adaptive bound optimization](https://arxiv.org/abs/1002.4908) for convex non-smooth functions, [multidimensional Armijo](https://arxiv.org/abs/2306.02527) for smooth strongly convex functions, or [parameter-free methods in online learning](Coin Betting and Parameter-Free Online Learning).


## Questions

- Please clarify what is meant by "maintaining constant memory usage" and "memory is constant with increasing training iteration". My understanding of the algorithm is that the methods learns a different preconditioner for each iteration. This scales at least linearly with the number of training iterations. A more detailled explaination of where the memory is used and how it differs from the unrolling strategy would help.

- I don't understand Eq. 16. What is $B_k$? Equation (14) treats both $G_\theta$ and $B_k^t(\tehta)$ as equations of $\theta$, which makes

## Minor

- Proposition 1 & 2 are missing the assumption that no entries of $\nabla f(x)$ is $0$. It is possible for the $j$th entry of $\nabla f(x)$ to be 0 while having $x[j] \neq x^*[j]$ where $x^* \in \arg\min_x f(x)$.

---

> ### Author Response · Authors · 2024-11-21
>
> We thank the reviewer for their detailed feedback.
>
> > ### “Would it be possible to guarantee that the algorithm would work on all $1/\tau$- smooth problems?”:
>
> We thank the reviewer for this comment and acknowledge that the current form of Theorem 2 may seem weak as we provide only the existence of some $\tilde{L}$, and do not explicitly show what this constant is.
>
> When checking the proof in detail, one observes that \tilde{L} is greater than the maximum Lipschitz constant in the training set. So the algorithm is in fact convergent on all $L_{\text{train}} = \max \{ L_1, \cdots L_N \} = 1/\tau$ - smooth functions given the theorem assumptions.
>
> We updated the statement of the theorem to  “then, there exists a final training iteration $T$ such that for all $f \in \mathcal{F}_{L  train}$ and any starting point $x_0$, using Algorithm 2, we have $\nabla f(x_t) \to 0$ as $t \to \infty$”.
>
> > ### “The proposed algorithm still relies on hand-crafted a-priori knowledge … it does not significantly differ from alternative approaches that require safeguarding or search within a predefined set that guarantees convergence.”:
>
> We agree that our approach utilizes a-priori knowledge to ensure convergence for generalization, but in contrast to safe-guarding it does so with soft constraints, rather than hard constraints.
>
> Moreover, exact knowledge of Lipschitz constants is not required. Any parameter that would make gradient descent convergent on training data is sufficient for our framework.
>
> Moreover, we believe that a key innovation of our method lies in the ability to learn parameters over hundreds or thousands of iterations, which is not seen in other L2O approaches.
>
> > ### “both problems are of the form $\| Ax - b \|^2$…  the linear operator $A$ is fixed. This seems like an ``easy'' problem”:
>
> You are correct that in the current experiments, the operator $A$ is fixed across training and test problems, and the variation in the function f only comes from the observation $b$. However, this is exactly the problem a practical translation would face: e.g. an imaging system (which defines a fixed A and data fit) scans dozens of patients every day ($b$ changes). The same setting is also considered in Banert et al 2024 https://epubs.siam.org/doi/epdf/10.1137/22M1532548.
>
> Note that the objective functions used in our numerical experiments aren’t just of the form $\|Ax-b\|^2$ but $f(x) = \|Ax-b\|^2 + \alpha H{\epsilon}(x)$, with $H{\epsilon}(x)$ a non-quadratic function, adding complexity to the optimization problem.
>
> > "Convolutional preconditioners":
>
> We accept that a comparison to non-learned convolutional preconditioners would strengthen our submission. Currently, we are working on this.
>
> > "I don't understand Eq. 16":
>
> We apologize for the confusion around Equation 16.
>
> Equation 16 just states that $G_{\theta} \nabla f_k(x_k^t)$ is linear in $\theta$. This means that there exists a linear operator $B_k^t$ such that $G_{\theta} \nabla f_k(x_k^t) = B_k^t \theta$.
>
> > "maintaining constant memory usage":
>
> Thank you for raising this point. The limitation with (a standard implementation of) unrolling is that the GPU needs to store all intermediate values to backpropagate, which scales memory with O(T). However, in our case, when the parameters \theta_t are learned and the next values $x_k^{t+1}$ are calculated, $\theta_t$ and $x_k^t$ are no longer required to be stored in the GPU and therefore can just be saved to disk, meaning that the GPU memory requirement scales as O(1) instead of O(T). This explanation will be added in the revised draft.
>
> We thank the reviewer for the other suggestions for clarification.

---

> > ### Author Response · Authors · 2024-12-01
> >
> > > theorem 2 both uses that $t \to \infty$ while requiring a final training iteration $T$. These two statements seem contradictory?
> >
> > One can replace the assumption $\lim\inf_t \lambda_t > 0$ in Theorems 2 and 3 with "there exists an iteration $T_0$, and a constant $\lambda > 0$ such that $\lambda_t \geq \lambda$ for all $t > T_0$". However, we aimed to write the Theorems as if we can consider an infinite number of training iterations as in Theorem 1, but then we can say there exists an iteration $T$ such that we obtain provable convergence on a class of unseen functions.
> >
> > > Proposition 1 & 2 are missing the assumption that no entries of $\nabla f(x)$ is $0$.
> >
> > We are considering convex functions. Proposition 4 (Appendix Section C) reads: "Assume that $f: \mathcal{X} \to \mathbb{R}$ is convex, continuously differentiable, and has a global minimum. Then for a point $z \in \mathcal{X}$ if there exists some $x^* \in \arg\min_x f(x)$ such that $[z]_i = [x^*]_i$, then $[\nabla f(z)]_i = 0$."
> >
> >
> > Thank you again for your helpful comments.

---

### Official Review · Reviewer_tgvj · 2024-11-03

**Soundness:** 4
**Presentation:** 3
**Contribution:** 3
**Rating:** 6
**Confidence:** 4

**Summary:**

The paper proposes to use greedy learning to help scale L2O methods by avoiding memory constraints of unrolling, allowing to train for more iterations. The paper focuses on learning a linear operator preconditioning operator, minimizing the function value at subsequent iteration. The paper shows that such parametrisation, by virtue of generalising gradient descent, the iterations admit provable convergence guarantees, even on unseen data. Such preconditioner parametrization is shown to outperform classical optimization algorithms, such as NAG and L-BFGS, in experiments on two image inverse problems: image deblurring and Computed Tomography.

**Strengths:**

The theoretical part of the paper is well-written and provides a very clear story. The theoretical framework also provides a good starting point for further generalising analysis of L2O schemes, especially when moving to non-convex settings.

The proposed preconditioner learning is memory-efficient, fast, with convergence guarantees and empirical evidence on small problems.

**Weaknesses:**

Major:

Main weaknesses appear in the numerical section of this paper. Overall, the numerical section is very difficult to read, as it seems to mix the exact parameter choices with the rest of the explanations, further obfuscating everything. The various details seemed to have been mixed into a single soup of information - this should be summarised better.

The numerical comparison seems to be missing the main evaluation - it is unclear whether the method actually generalises. Only a small example is portrayed, and no evaluation over the whole dataset is provided. It is not clear to me whether Figure 2 converges and Figure 6 diverges simply due to a different example being provided. Preferably there would also be some analysis of the initialisation question for the problems of consideration.

There is also a summary/interpretation missing from the numerical section - it is not clear what the numerical results seem to be showing beyond efficiency over classical methods? Why are fully learned preconditioners bad? Is this observed for all problems?

There also seems to be very little mentioned about the limitations of this approach - can this be expanded upon? Currently (at least to me), the question of computational costs behind hyperparameter tuning is not very clear. Also, memory footprint is unclear to me - the greedy approach needs to store a matrix for each step - thus linearly increasing memory requriement with T?

Minor:

The method has only been illustrated on rather small scale problems (limited to 40x40 and 96x96) - it would be interesting to see whether same behaviour is observed on more realistic image sizes like 256x256 or 1024x1024.

The paper is limited to convex problems, and while this is a good starting point I do believe that this is a significant weakness, especially given the interest in using L2O for optimization of non-convex problems. In the same vain, this approach (or at least the analysis) seems to be limited to differentiable functions.

**Questions:**

* Equation 2 seems to only vary the y over the whole space. I believe this should be rewritten to emphasise what kind of functions you expect to be varying over. I.e. y should be from some underlying distribution? Do regularisers get varied? Do operators get varied?
* Line 108, you seem to choose X to be a finite dim Hilbert space - what is the value of this? If X is finite dim, then practically there is not point in distinguishing it from Euclidean space or am I missing something?
* Proposition 2 - linear independence seems like a rather arbitrary assumption. Can anything be said when it does not hold?
* Equation 9 - this is a different regularizer from the one in equation 2, so worth using different notation for the two.
* Theorem 3 seems to assume that $x_t$ has to now be a bounded sequence, which seems to be a relatively strange assumption to appear in such a context. Can you explain why this is necessary?
* Random question to authors: do you believe that L2O can overcome the problems discussed in https://arxiv.org/pdf/2301.06148?
* Section 6 - in this section what is Y?
* Figure 3a - this is not reconstruction, but initialization presumably?
* Figure 5 - where is the ground truth? Why is it only sinogram and reconstruction shown?

---

> ### Author Response · Authors · 2024-11-22
>
> We thank the reviewer for their detailed constructive feedback.
>
> > ### “No evaluation over the whole dataset is provided.”:
>
> We are sorry that our writing wasn't clear enough. The plots in Figure 2, Figure 4 and Figure 6 (and Tables 3 and 4 in the appendix) use the function F which is the mean function value over the entire data set (training or test, respectively) as defined in the paragraph starting in line 329.
>
> In the revision we will add The maximum and minimum function value vs iteration for each method too to make this more clear.
>
> > ### “memory footprint is unclear to me”:
>
> Thank you for raising this point. The limitation with (a standard implementation of) unrolling is that the GPU needs to store all intermediate values to backpropagate, which scales memory with O(T). However, in our case, when the parameters \theta_t are learned and the next values $x_k^{t+1}$ are calculated, $\theta_t$ and $x_k^t$ are no longer required to be stored in the GPU and therefore can just be saved to disk, meaning that the GPU memory requirement scales as O(1) instead of O(T).
>
>
> > ### “It would be interesting to see whether same behaviour is observed on more realistic image size.”:
>
> Thank you for the feedback. A numerical experiment will be added to the paper with a 256 x 256 image size for the CT problem. For this problem, we observe very similar results to the two experiments shown in the first submission. In particular, we see that the learned algorithm using a convolutional parametrization outperforms L-BFGS and NAG on test data.
>
> > ### Response to Questions
>
> > “Equation 2 seems to only vary the y over the whole space. I believe this should be rewritten to emphasise what kind of functions you expect to be varying over.”:
>
> Thank you for noticing this, you are correct and this has been changed in the revision.
>
> Furthermore, the regularizer and the operator $A$ are fixed across training and test problems, so the variation in the function f only comes from the observation $b$. However, this is exactly the problem a practical translation would face: e.g. an imaging system (which defines a fixed A and data fit) scans dozens of patients every day ($b$ changes). The same setting is also considered in Banert et al 2024 https://epubs.siam.org/doi/epdf/10.1137/22M1532548.
>
> > “Proposition 2 - linear independence seems like a rather arbitrary assumption.”:
>
> You are correct that it seems an arbitrary assumption. One can obtain the same result with a more general assumption, which will be included in the revised submission.
>
> > “Theorem 3 seems to assume that $x_t$ has to now be a bounded sequence, which seems to be a relatively strange assumption to appear in such a context.”
>
> The assumption of a bounded sequence is used to ensure the convergence of our method in the greedy setting, see line 936-941, it is used to bound f(x_t) - f(x^*) in terms of \| \nabla f (x_t) \|.
>
> > “Section 6 - in this section what is Y?”:
>
> Thank you for this question, $\mathcal{Y}$ is never explicitly detailed in section 6. It is the observation space, e.g. for the deblurring case $\mathcal{Y} = \mathcal{X}$. This will be added in the revised paper.
>
> > “Random question to authors: do you believe that L2O can overcome the problems discussed in https://arxiv.org/pdf/2301.06148?”:
>
> Thank you for the question, however we are unable to answer this currently. We will look into this further.
>
> > “Line 108, you seem to choose X to be a finite dim Hilbert space - what is the value of this?”
>
> Yes, you are correct that we can consider Euclidean spaces. Often one considers Hilbert spaces in optimization.
>
> Lastly, we thank the reviewer for noticing inconsistencies in Equation 9 and Figure 3a.

---

> > ### Comment · Reviewer_tgvj · 2024-11-23
> >
> > I would like to thank the authors for their extensive response. Overall, most of the questions raised have been answered and I feel confident in raising my score to an accept, as long as the numerical section is improved  (which I mentioned to be a major weakness previously) - I look forward to seeing the updated manuscript.
> >
> > There was only one questions I would still like to get a response to:
> > "There also seems to be very little mentioned about the limitations of this approach - can this be expanded upon? Currently (at least to me), the question of computational costs behind hyperparameter tuning is not very clear."

---

> > > ### Author Response · Authors · 2024-11-24
> > >
> > > We thank the reviewer for their continued feedback.
> > >
> > > > ### Limitations:
> > >
> > > A limitation of our work is the requirement of knowledge about the Lipschitz-smoothness constants of training functions $f_k$. However, we only need to know a step size that would lead to convergence with gradient descent for all training functions. For example, one can use any upper bound of the Lipschitz-smoothness constants of functions. It is worth noting that the knowledge of the Lipschitz-smoothness is also considered in other L2O works, for example, Banert et al 2024 https://epubs.siam.org/doi/epdf/10.1137/22M1532548.
> > >
> > > Furthermore, this work is restricted to only convex functions as this allows us to learn parameters $\theta_t$ that globally optimize the function $g_{t, \theta_t}$. However, convex functions are of course an important class within optimization, particularly in the field of inverse problems.
> > >
> > >
> > > > ### Hyperparameters:
> > >
> > > When calculating parameters $\theta_t$, using our approach we calculate the Lipschitz-smoothness $L_{g_{t, \lambda_t}}$ of the convex objective function $g_{t, \lambda_t}$ and then use Nesterov’s Accelerated Gradient method with step size $1/L_t$, which requires no hyperparameter tuning.
> > >
> > > Often when calculating parameters for L2O, one has to tune the learning rate (and any other hyperparameters of the algorithm used, e.g. Adam).
> > >
> > > Please let us know if you require any further clarification.

---

> > > > ### Author Response · Authors · 2024-12-01
> > > >
> > > > > Proposition 2 - linear independence seems like a rather arbitrary assumption. Can anything be said when it does not hold?
> > > >
> > > > We decided to not include a more general version of the result in the revised paper, since we wanted to show that the result can hold for a simple and easily interpretable assumption. However, the assumption may be generalised by requiring that $\operatorname{Range(B)} \subseteq \operatorname{Range(A)}$ for matrices $A = \begin{bmatrix}
> > > >          \nabla f_1(x_1^0) | \cdots | \nabla f_N(x_N^0)
> > > >     \end{bmatrix}, B = \begin{bmatrix}
> > > >          x_1^0 - x_1^* | \cdots | x_N^0 - x_N^*
> > > >     \end{bmatrix}$.
> > > >
> > > > Thank you again for your helpful comments.

---

### Author Response · Authors · 2024-12-01
**Paper Revision**

Thank you to all the reviewers for carefully reading our submission and for your thoughtful and constructive feedback.

In particular, we have made updates to the numerical experiments section. Here are the major changes:
* We have added a comparison to a hand-crafted convolutional algorithm for the deblurring problem.
* We have added a CT problem on 256x256 images to benchmark on a larger-scale example.
* We have added maximum and minimum values over the dataset to show the best and worst performance of the learned convolutional algorithm over the test dataset.
* In addition to the objective function value against iteration number, we have added plots of objective value against wall clock time, demonstrating that our learned convolutional algorithm also outperforms classical hand-crafted optimizers with respect to time.

---

### Meta-Review · Area_Chair_jar4 · 2024-12-29

**Metareview:**

The paper studies a greedy training method for learning to optimize. In the proposed approach, parameters are determined sequentially. Given the parameters for the first k iterations, parameters for iteration k+1 are chosen to minimize the average objective value over the training set. Greedy training has efficiency advantages compared (e.g.) to loop unrolling, since one doesn’t have to backpropagate through multiple iterations. For linear parameterizations (i.e., a step size, element wise scale, or preconditioning matrix) this parameter selection operation is a convex program (indeed, simply least squares if the objective is the squared error).

The paper analyzes the greedy scheme theoretically, arguing that (1) if the class of admissible step rules includes gradient rules, then greedy learning performs at least as well as gradient decent on training data, and (2) if the regularization parameters are chosen such that for large iterations, the chosen step rule tends to the gradient rule, then the performance on unseen (test) data inherits the convergence rate of gradient descent. The proposed approach is applied to inverse problems in image deblurring and computed tomography, where it used to learn convolutional preconditioners.

The main strength of the paper is its relatively simple, practical proposal for greedy learning to optimize. Compared to unrolling approaches, this method is scalable to large numbers of iterations without needing to backpropagate, and with constant memory. As described below, the discussion clarified a number of issues around the paper’s theory — in particular the meaning of the “better than gradient descent” condition. At the same time, reviewers retained concerns about the paper’s generalization theory and its experiments, which would be stronger with comparisons across problem types (e.g., different blur operators) and with comparisons to existing approaches to L2O (not just classical optimization baselines).

**Additional Comments On Reviewer Discussion:**

Reviewers generally found the proposed greedy approach to L2O to be an effective approach to controlling the complexity of training. Reviewers raised the following issues during the discussion.

- Meaning of the BGD (better than gradient descent) assumption [aeaw, yL8M]. This issue was well-clarified by the discussion: BGD simply requires that average function value on the training problems is no larger than that achieved by gradient descent, which is guaranteed as long as search space includes gradient descent. Questions were also raised about the strength of the guarantee of generalization to unseen data.
- Comparison with existing schemes for L2O [yL8M] and across problem types [aeaw] and parameters for classical methods [6jvT] - in particular, state of the art convergent methods for L2O [yL8M].
- Safeguarding: the method requires to know the maximum Lipschitz constant over the entire training set [aeaw,6jvT]

While the author response addressed some reviewer concerns, especially around the meaning of the paper’s theoretical assumptions, reviewer evaluation remained mixed.

---

### Decision · Program_Chairs · 2025-01-22

Reject